# Interpretability of negative latent heat fluxes from Eddy Covariance measurements during dry conditions

Sinikka J. Paulus[1, 2, 3], Rene Orth[1, 3], Sung-Ching Lee[1], Anke Hildebrandt[4, 2], Martin Jung[1], Jacob A. Nelson[1], Tarek Sebastian El-Madany[1], Arnaud Carrara[5], Gerardo Moreno[6], Matthias Mauder[7], Jannis Groh[8,9,10], Alexander Graf[9], Markus Reichstein[1], and Mirco Migliavacca[1, 11]

[1]Max Planck Institute for Biogeochemistry, Department of Biogeochemical Integration, Jena, Germany
[2]Institute of Geoscience, Friedrich Schiller University, Jena, Germany
[3]Faculty of Environment and Natural Resources, University of Freiburg, Germany
[4]Department Computational Hydrosystems, Helmholtz Centre for Environmental Research (UFZ), Leipzig, Germany
[5]Fundacion Centro de Estudios Ambientales del Mediterráneo (CEAM), Valencia, Spain
[6]Institute for Silvopastoralism Research (INDEHESA), Universidad de Extremadura, Plasencia , Spain
[7]Technische Universität Dresden, Dresden, Germany
[8]Institute of Crop Science and Resource Conservation (INRES) - Soil Science and Soil Ecology, University of Bonn, Bonn, Germany
[9]Institute of Bio- and Geoscience IBG-3: Agrosphere, Forschungszentrum Jülich, Jülich, Germany
[10]Research Area 1 Landscape Functioning, Isotope Biogeochemistry and Gas Fluxes, Leibniz Centre for Agricultural Landscape Research (ZALF), Müncheberg, Germany
[11]current address: European Commission, Joint Research Centre, Ispra, Varese

**Correspondence:** Sinikka J. Paulus (spaulus@bgc-jena.mpg.de)

**Abstract.** It is known from arid and semi-arid ecosystems that atmospheric water vapor can directly be adsorbed by the soil matrix. Soil water vapor adsorption was typically neglected and only recently got attention because of improvements in measurement techniques. One technique rarely explored for the measurement of soil water vapor adsorption is eddy covariance (EC). Soil water vapor adsorption may be detectable as downwards-directed (i.e., negative) EC latent heat ($\lambda E$) flux measurements under dry conditions, but a systematic assessment of the use of negative $\lambda E$ fluxes from EC flux station to characterize adsorption is missing. We propose a classification method to characterise soil water vapor adsorption, while excluding conditions of dew and fog when $\lambda E$ derived from EC is not trustworthy due to stable atmospheric conditions. We compare downwards-directed $\lambda E$ fluxes from EC with measurements from weighable lysimeters for four years in a Mediterranean Savannah ecosystem and three years in a temperate agricultural site. Our aim is to assess if overnight water inputs from soil water vapor adsorption differ between ecosystems and how well they are detectable by EC.

At the Mediterranean site, the lysimeters measured soil water vapor adsorption each summer whereas at the temperate site soil water vapor adsorption was much rarer, and measured predominantly under an extreme drought event in 2018. In 30 % of nights in the four-year measurement period at the Mediterranean site, the EC technique detected downward-directed $\lambda E$ fluxes of which 88.8 % were confirmed to be soil water vapor adsorption by at least one lysimeter. At the temperate site, downward-directed $\lambda E$ fluxes were only recorded during 15 % of the nights, with only 36.8 % of half-hours matching simultaneous lysimeter measurement of soil water vapor adsorption. This relationship slightly improved to 61 % under bare soil conditions and extreme droughts. This underlines that soil water vapor adsorption is likely a much more relevant process in

arid ecosystems compared to temperate ones and that the EC method was able to capture this difference. The comparisons of the amounts of soil water vapor adsorption between the two methods revealed a substantial underestimation of the EC com-

pared to the lysimeters. This underestimation was, however, comparable with the underestimation in evaporation by the eddy covariance and improved in conditions of higher turbulence. Based on a random forest-based feature selection we found the mismatch between the methods being dominantly related to the site's inherent variability in soil conditions, namely soil water status, and soil (surface) temperature.

We further demonstrate that although the water flux is very small with mean values of 0.04 or 0.06 mm per night for EC or

lysimeter, respectively, it can be a substantial fraction of the diel soil water balance under dry conditions. Although the two instruments substantially differ with regard to the measured ratio of adsorption over evaporation over 24 hours with 64 % and 25 % for the lysimeter and EC methods, they are in either case substantial. Given the usefulness of EC for detecting soil water vapor adsorption as demonstrated here, there is potential for investigating adsorption in more climate regions thanks to the greater abundance of EC measurements compared to lysimeter observations.

**1   Introduction**

The adsorption of atmospheric water vapor by dry soils (**S**oil **V**apor **A**dsorption - SVA) has in recent years been identified to be underrepresented in ecosystem research (Saaltink et al., 2020). When the volumetric soil water content ($SWC$, m$^3$ m$^{-3}$) is low, water molecules are bound stronger in the liquid phase. As a result, the balance between the liquid and vapor phases shifts, leading to a reduction in the relative humidity ($rH$, %) within the air-filled pore space of the soil. Consequently, under

such soil hydraulic conditions the soil can effectively act as a sink of atmospheric vapor.

Although the adsorption of water vapor on soil particles has a long history of research (e.g. Hansen, 1926; Orchiston, 1953; Philip and De Vries, 1957; Edlefsen et al., 1943; Tuller et al., 1999), and many theoretical and empirical models exist to describe it mathematically (Arthur et al., 2016), little is known about the extent and relevance of SVA in ecosystems (for the theoretical background of the process, see section 2).

Measurements of SVA in natural and managed ecosystems with the perspective to quantify its role as a water input have traditionally been performed using cloth plates (Kidron, 1998), weighable lysimeters (Kidron and Starinsky, 2019; Verhoef et al., 2006; Uclés et al., 2013; Feigenwinter et al., 2020; Paulus et al., 2022), and sampling campaigns (McHugh et al., 2015). Although uncertainties can emerge due to temperature differences between the (micro)-lysimeter and the surrounding soil (Kidron and Kronenfeld, 2020) when temperature control is lacking, the latest generation of large high-precision weighing

lysimeters now features sensor arrays. These sensor arrays enable the measurement of soil variables both inside and outside the lysimeter column, enabling the monitoring and control of boundary conditions very similar to those in the undisturbed soil environment (Pütz et al., 2018). Model-based numerical evaluations further confirmed the ability of this type of lysimeter to correctly quantify SVA (Saaltink et al., 2020). Based on the analysis of long time series, SVA was observed to reach significant magnitudes. For example, in one coastal dune it was estimated to be 77 $\mathrm{kg\,m^{-2}\,y^{-1}}$ (Saaltink et al., 2020). In another case

study in a semi-arid region, SVA accounted for up to 40 % of diel evaporation during the crop growth period (Zhang et al.,

2019). Furthermore, in a Mediterranean tree-grass ecosystem, SVA served as the sole water input for several consecutive weeks in the dry season (Paulus et al., 2022).

While these findings provide valuable insights into the importance of this flux and improve the temporal coverage, they also highlight the existing knowledge gap when it comes to spatial representation. This gap arises primarily due to limitations in measurement techniques, as current methods predominantly rely on the aforementioned large weighing lysimeters, which require substantial investment and maintenance. As a consequence, alternative approaches for measuring SVA have been developed. These include the gradient method (Lopez-Canfin et al., 2022), the utilization of soil chambers (Qubaja et al., 2020), and the application of relative humidity sensors in the soil (Kool et al., 2021). These techniques share the common goal of finding alternative means of measuring SVA, aiming to enhance data coverage and improving our understanding of this process.

Previous studies reported simultaneous measurements of downward (negative) latent heat fluxes ($\lambda E$, $\mathrm{W\,m^{-2}}$) using the Eddy Covariance (EC) method alongside independent SVA measurements (Qubaja et al., 2020; Paulus et al., 2022). Florentin and Agam (2017) compared SVA from an EC measurement system with microlysimeter measurements over a 7-day period in the Negev desert and found that while the EC method accurately captured the dynamics of SVA, it did not fully capture its magnitude. In theory, EC should be able to measure SVA at the ecosystem scale. However, negative $\lambda E$ fluxes measured by the EC are rather small and have generally been regarded as random noise and, in some cases, disregarded altogether.

Weighable lysimeters and EC are both standard techniques used to measure evaporation *in situ* but the measurement principles differ substantially. In this manuscript we will use the umbrella term evaporation for all vapor fluxes at the land surface, in accordance with (Miralles et al., 2020); as we are mainly concentrating on periods with little or no vegetation activity. The weighing lysimeter method is based on changes in the weight of the lysimeter, which are assumed to be caused exclusively by changes in the amount of water within the measurement volume. The EC method is based on the covariance between vertical wind speed and vapor density, from which $\lambda E$ is calculated. EC provides a high spatial (from few hundred squared meters) and temporal resolution measurements of water fluxes at relatively low operating costs compared to weighable lysimeters but the method carries many uncertainties introduced by low atmospheric turbulence, sensor maintenance, and data processing (Mauder et al., 2013). In addition, EC measures the turbulent vertical transport of gases at a few meters above the soil surface whereas lysimeters measure the phase change of water (vapor $\rightleftharpoons$ liquid or solid) at the ground level. Another difference between lysimeters and EC is that the size and shape, and position of the surface area of influence varies for EC depending on the wind speed and direction and turbulence conditions (Amiro, 1998; Schmid, 1994, 2002), whereas lysimeters are spatially stationary and always measure the same volume of soil. Several comparisons exist between those instruments for evaporation (Gebler et al., 2015; Hirschi et al., 2017; Mauder et al., 2018) and it has been found that EC underestimates evaporation fluxes under conditions of low friction velocity. Less work has focused on comparing non-rainfall water inputs (i.e., SVA, dew, and fog), but it has been reported that EC systems suffer from inaccuracies in flux measurements under conditions of high $rH$ (Fratini et al., 2012; Zhang et al., 2023) and stable atmospheric stratification, which limits their ability to measure dew formation (Moro et al., 2007; de Roode et al., 2010) and fog deposition (Eugster et al., 2006; El-Madany et al., 2013). However, SVA does not dependent on atmospheric stability. SVA can occur at relatively low $rH$ levels and high surface temperatures ($T_s$, °C). Therefore, compared to dew and fog, EC measurements should be more accurate for SVA.

Research on SVA has mainly focused on dry regions, where the movement of water vapor into the upper soil is significant due to consistently low SWC. While SVA has been observed in temperate climates during late summer in uncovered, dry soils (Blume et al., 2016a), it is likely to be much less relevant due to the overall higher SWC.

The use of EC to detect and quantify SVA would be particularly beneficial given the availability of global long-term observatory networks (e.g. FLUXNET) (Baldocchi et al., 2001). Analyzing existing EC data series could immediately significantly improve our understanding of SVA, at both spatial and temporal scales. However, the potential and limitations of the EC technique for measuring SVA need to be assessed. In this study, we investigate the potential of EC to measure SVA. We hypothesize that the effect of the soil matrix to adsorb water molecules under dry conditions i) is higher in the Mediterranean than in the temperate climate, ii) these differences in SVA can be detected by EC, and iii) SVA can be quantified by EC despite the vertical distance between the EC sensors and the adsorbing soil surface and despite the measurement uncertainties resulting from low nighttime turbulence and random noise. We use co-located lysimeters and EC measurement stations to test our hypothesis assuming that the median lysimeter signal is the ground truth, representing field heterogeneity.

## 2    Theoretical background on soil water retention

"Water vapor adsorption refers to the influx of water vapor from the atmosphere into a soil followed by condensation. It involves vapor diffusion and water retention [...]." (c.f. Saaltink et al., 2020). Water in the soil is subject to several forces and their combined effect is expressed as the deviation of potential energy of the soil water relative to the reference state. The difference in chemical and mechanical potentials between soil water and pure water at the same temperature is defined as the soil water potential($\Psi_w$, hPa) and is generally expressed in units of pressure. Although the more more widely *in situ* measured volumetric $SWC$ and $\Psi_w$ are linked, in contrast to $SWC$, $\Psi_w$ describes the energy requirements to change the phase state of water or to induce water transport. Therefore, at the same $SWC$, $\Psi_w$ can differ by an order of magnitude due to variations in soil physical properties (Or et al., 2022). The dominant force of the $\Psi_w$ is the matric potential ($\Psi_m$, hPa). $\Psi_m$ is a result of the combined effect of capillary and adsorptive forces (Tuller et al., 1999). One consequence of adsorptive forces under dry conditions is that fewer water molecules "escape" the liquid phase into the ambient atmosphere resulting in lower $rH$ (lower relative vapor pressure) in the air-filled pore space of the soil.

The vapor pressure above water at a reference state is, therefore, higher relative to the water held in soil pores by matric forces. This relationship is described by the Kelvin equation (Edlefsen et al., 1943) (given in Appendix B) and is key for the occurrence of SVA in ecosystems. Figure 1 illustrates this relationship in dry and wet soil conditions. We used the water retention curve of a typical loamy sand to derive $\Psi_w$ from $SWC$ (van Genuchten, 1980). In this example, we assume idealized conditions of an equilibrated system with a homogeneous temperature of 20 °C and constant atmospheric $rH$ of 60 %. During wet soil conditions (a) the pore vapor pressure is near saturation (100 % $rH$) and water evaporates and diffuses into the atmosphere. During dry soil hydraulic conditions (b) the equilibrium between the liquid and vapor phase is lower relative to the reference state: due to the low $\Psi_w$ water molecules already in the soil solution are prevented from "escaping" into the

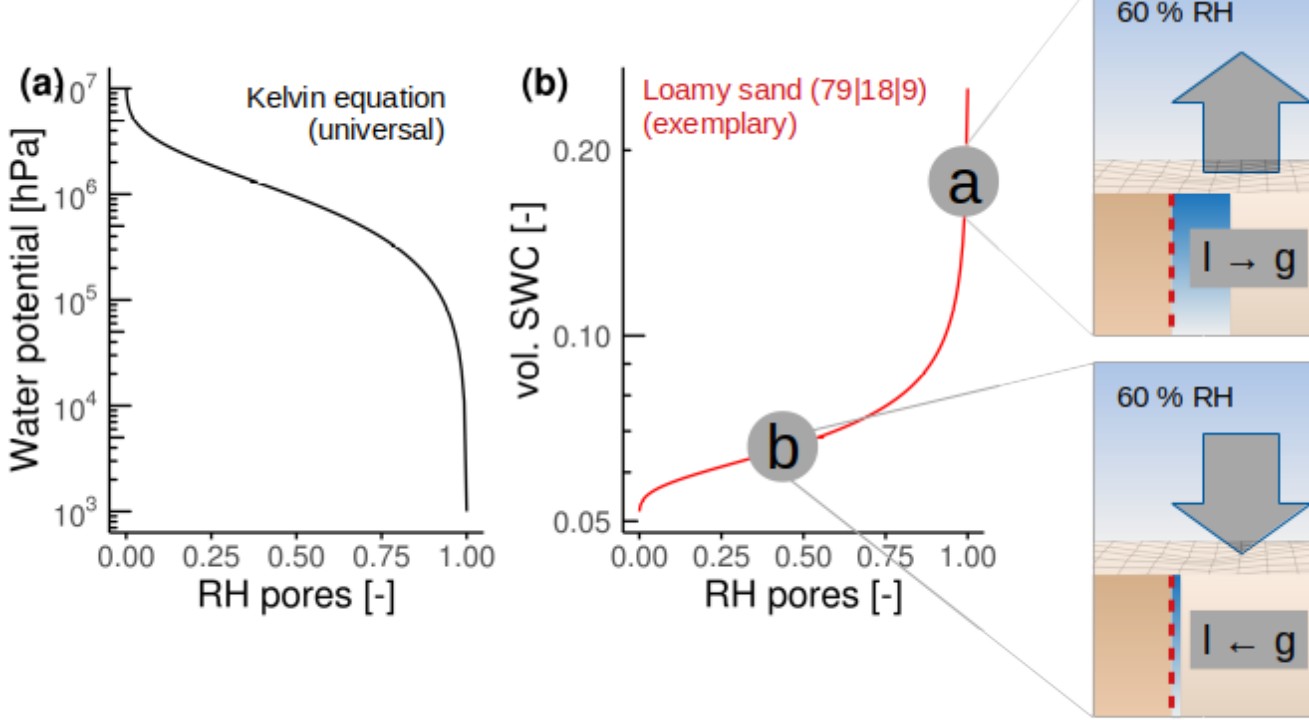

**Figure 1.** Relationship between (a) soil water potential and relative humidity (rH) of the soil pores at 20 degC defined with the Kelvin equation. (b) Illustration of the conversion of water potential from (a) to the respective volumetric soil water content (SWC, $m^3$ $m^{-3}$) for a loamy sand consisting of 79 % sand, 18 % silt and 9 % clay, based on the van Genuchten Model (van Genuchten, 1980). The representations (a and b) illustrate that at constant atmospheric rH of 60 % at a temperature of 20 °C, the vapor flux direction and phase change (l and g for liquid and gas) within the soil are opposite for different soil water potentials.

atmosphere and molecules entering the soil from the relatively wet atmosphere (60 % $rH$) are adsorbed onto the soil particles, maintaining a vapor concentration gradient from the atmosphere into the soil until the system equilibrates.

Due to non-equilibrated conditions and spatiotemporal temperature variations, the processes under natural conditions are much more complex than in this example. But since adsorptive forces are intrinsic soil physical properties, the adsorption of atmospheric vapor can theoretically occur in any ecosystem on condition that the soil is dry enough, the atmosphere carries enough moisture, and the boundary conditions for vapor transport (aerodynamic resistance) allow vapor flow into the soil.

## 3 Material and Methods

 ### 3.1 Site descriptions

The study was conducted at the experimental field sites Majadas de Tiétar, Extremadura, Spain ($39°56'25.12''N05°46'28.70''E$; 260 m asl, ES-LMa*) and Selhausen, Lower Rhine Valley, Germany ($50°52'7''N06°26'58''E$; about 103 m asl., DE-RuS).

**Majadas de Tiétar (ES-LMa*)**: The field site is a Mediterranean (summer-dry) tree-grass ecosystem. The nearest sea is the Atlantic 272 km to the west. The site experiences an average annual temperature of 16.7 °C and receives approximately 650 mm of rain annually over 2004-2022, primarily falling between November and May, followed by extended dry summers (El-Madany et al., 2018). The vegetation at the site is characterized by a sparse tree cover of about 20 %, mainly consisting of *Quercus ilex (L.)* with an approximate density of 20 trees per hectare (Bogdanovich et al., 2021), and pasture understory regularly grazed by cattle. During the growing season, the herbaceous layer dominates, comprising grasses, forbs, and legumes. The fractional cover of these plant forms varies seasonally based on their phenological stage, with important interannual variations influenced by the precipitation seasonal distribution (Perez-Priego et al., 2017). The herbaceous layer typically reaches its peak in late March, with a mean plant area index of up to about 2 $m^2$ $m^{-2}$, undergoes senescence by the end of May, and regains its greenness about October (Migliavacca et al., 2017). The soil at the site is classified as an Abruptic Luvisol (Ah, Bt, Btg, C). The upper horizons are characterized according to the USDA classification system as loamy sand (75 % sand, 5 % clay, and 20 % silt) sitting on top of a clay horizon (52 % sand, 18 % clay, and 30 % silt) which starts at a depth of 30 to 100 cm (U.S. Department of Agriculture, 2017; Nair et al., 2019). The regional clay mineralogy was identified as a blend of smectite (45 %), illite (35 %) and chlorite and/or kaolinite (20 %) (NC Geological Survey of Spain (IGME), 1992).

**Selhausen (DE-RuS)**: The agricultural research site, Selhausen, is part of the TERENO-Rur hydrological observatory (Bogena et al., 2018) and contains a lysimeter station and an EC flux tower, which are part of the TERENO-SOILCan lysimeter network in Germany (Pütz et al., 2016) and the Integrated Carbon Observatory System (ICOS) (Integrated Carbon Observation System Heiskanen et al., 2022). The site consists of 51 agricultural fields (with a total area of  1 $km^2$) representing the heterogeneous rural area in the Lower Rhine Valley. It belongs to the temperate maritime climate zone, with a mean annual temperature of 10.2°C and with 714 mm of annual precipitation uniformly distributed over the year (Bogena et al., 2018). The site is agriculturally managed with rotating crops (winter wheat, winter barley, winter rye, potato, oat, and catch crops) during the period of investigation, with a winter cereal-only rotation on the lysimeters. As a consequence of the tillage, seeding, and harvest activities, there are large inter-annual variations in the thickness of the vegetation layer, including prolonged periods of bare soil. The soil at the site is classified as a Cutanic Luvisoll (Pütz et al., 2016) and the soil texture of the different soil horizons (Ap, Al-Bv, II-Btv) can be classified according to USDA 2017 as silt loam (U.S. Department of Agriculture, 2017; Groh et al., 2020). The clay mineralogy of the site was identified as a blend predominantly illite with the presence of chlorite and/or vermiculite and little amounts of kaolinite (Jiang et al., 2014)

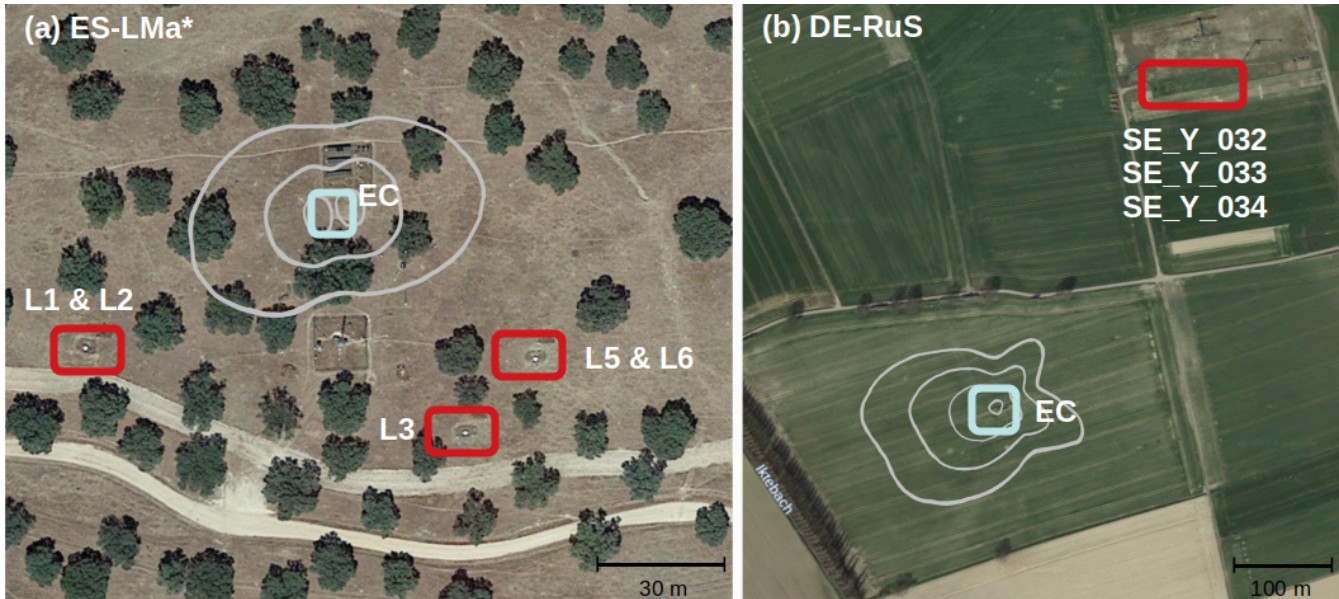

**Figure 2.** Aerial image of (a) the Majadas de Tiétar (ES-LMa*) and (b) the Selhausen agricultural field site (DE-RuS). The squares show the location of Eddy Covariance (EC) instruments (light blue) and the lysimeters (red) at each site. The EC footprint climatology isolines are overlaid in grey (for 50 %, 70 %, and 80 % of the climatology, respectively; Selhausen: based on ICOS(2021)). Note that the spatial resolution differs. (Map data from Google Earth; (a) image from Instituto Geográfico Nacional, (b) image from GeoBasis-DE/BKG).

Aerial pictures of both sites with the associated footprint climatology of the eddy covariance measurements are shown in Figure 2.

## 3.2 Eddy covariance and lysimeter measurements

At ES-LMa*, the EC system consists of a sonic anemometer (SA-Gill R3-50; Gill Instruments Limited, Lymington, UK), an enclosed path IR gas analyzer (LI-7200, LI-COR Biosciences Inc., Lincoln, NE, USA). It is located in an open area at a height of 1.6 m above ground to measure only the fluxes from the sub-canopy herbaceous layer. To avoid confusion with the whole ecosystem EC system located at 15 m height, we added an asterisk to the site ID. EC raw data were collected at 20 Hz and flux calculations were performed with EddyPro software (version 6.2.0.). Raw time series were first subjected to de-spiking and block-average means were then subtracted (Vickers and Mahrt, 1997). Coordinate rotation was performed using the planar fit method for the two primary wind directions (Wilczak et al., 2001), followed by the double rotation method for the remaining data. Standard integral turbulence characteristics were identified and most problematic records removed (Foken and Wichura, 1996). For more details about the setup and the processing please be referred to Perez-Priego et al. (2017) and El-Madany et al. (2018, 2020). The two dominant wind directions at ES-LMa* are East and West-South-West (Fig. 2a).

In DE-RuS the EC equipment of the DE-RuS station consists of a sonic anemometer (CSAT3, Campbell Scientific, Logan, UT, USA) and an open-path IR gas analyzer (LI-7500, LI-COR Biosciences Inc., Lincoln, NE, USA). Measurement height was 2.34 to 2.55 m above the soil surface near the center of a 9.8 ha crop field. EC raw data were collected at 20 Hz and flux calculations were performed with TK3.11 (Mauder et al., 2013). Raw time series were first subjected to de-spiking and block-average means were then subtracted. The planar fit method was performed uniformly across all wind directions. Data points not meeting the assumptions on stationarity and integral turbulence characteristics were removed (Foken and Wichura, 1996). More details about the site, instrumentation, and processing can be found in Ney and Graf (2018). The two softwares used to process the raw data at the two sites (EddyPro and TK3) have been shown to be in good agreement (Fratini and Mauder, 2014).

2D-Footprint analysis aiming to evaluate whether half-hourly flux values are sufficiently representative of the target area, were performed for both sites based on the model by Kljun et al. (2015) (illustrated as footprint climatology isolines in Fig. 2; ES-LMa: from 2015-2017; DE-RuS: 2018-2019, more details given in ICOS (2021)). At ES-LMa* the 80 % footprint climatology is within a distance of 33 m from the tower in the two dominant wind directions. At DE-RuS, 80 % of footprint climatology is within the agricultural field in the dominantly prevailing west-south-west wind direction. The conversion of $\lambda E$ was converted to water flux (mm) by dividing it by the latent heat of vaporization $\lambda$ ($\lambda = (2.501 - 0.00237 \times T_a) \times 10^{-6} \, \mathrm{J \, kg^{-1}}$). The energy imbalance for EC was calculated as the sum of half-hourly turbulent fluxes (H + LE) versus available energy (Rn - G). Note, that this leads to an overestimation due to the neglect of storage terms. The full EC time series from ES-LMa* and DE-RuS comprise eight years of data, each from 1 January 2015 to 31 December 2022.

The lysimeter measurement facility in ES-LMa* consists of three stations in three locations with a distance of 104, 91, and 24 m of each other, and a distance of 66, 56 and 55 m to the EC setup respectively (Fig. 2 a). Each station contains two weighable, high-precision, high density polyethylene lysimeters (Umwelt-Geräte Technik GmbH, Müncheberg, Germany) with a surface area of 1 m$^2$ surface area, and 1.2 m column height, each. The weight of each lysimeter column is measured with three precision shear stress load cells (model 3510, Stainless Steel Shear Beam Load Cell; VPG Transducer, Heilbronn, Germany, 0.01 kg measurement precision) at a temporal resolution of 1 minute. The lysimeters were installed in 2015 by excavating undisturbed soil monoliths from open grassland areas with the natural herbaceous vegetation being preserved. Each station has a lower boundary control system, consisting of a heat exchange system and porous ceramic bars at the bottom of each column to adjust soil temperature and water content to the conditions of the surrounding soil at the same depth (Groh et al., 2016; Podlasly and Schwärzel, 2013). More details on the technical specifications are described by Paulus et al. (2022) and the excavation method by Reth et al. (2021). Within each column, $SWC$ and soil temperature ($T_{soil}$, °C) (UMP-1, Umwelt-Geräte-Technik GmbH) are measured at 0.1 m soil depth at a resolution of 0.1 % SWC and 0.02 °C, according to the manufacturer. Heat dissipation sensors, also located at 0.1 m soil depth, additionally provide estimates of $\Psi_m$ (Tensiomark, EcoTech Umwelt-Messsysteme GmbH, Bonn, Germany). However, it should be noted that the suitability of the heat dissipation method is under debate and this sensor in particular was reported to yield inaccurate readings under dry conditions (Degré et al., 2017; Jackisch et al., 2018). We therefore use the readings only as an indicator of the spatial heterogeneity of $\Psi_m$ and do not interpret the

absolute readings. We calculated $rH$ and vapor pressure of the soil air ($e_{soil}$, hPa) from $\Psi_m$ and $T_{soil}$ with the Kelvin equation (Edlefsen et al., 1943) (given in Appendix B).

The lysimeter measurement facility in DE-RuS consists of 4 lysimeter stations, each hosting a set of 6 weighable lysimeters.
The 24 lysimeters were filled with eight different soil types (each soil 3 replications), however, for the comparison we use data from 3 lysimeters that contain the local soil from Selhausen (SE_Y_032, SE_Y_033, and SE_Y_034, https://www.tereno.net/), and exclude other soils that are part of the translocation experiment within TERENO-SOILCan (Pütz et al., 2016). The lysimeters in Selhausen are arranged hexagonally (six lysimeters per station), with a distance of about 1.2 m between two adjacent lysimeters. This comprises three weighable high-precision, stainless-steel lysimeter columns (UMS AG, München, Germany) (Fig. 2b). Please note that there is a distance of 357 m between lysimeters and the EC set up at DE-RuS, and the agricultural management deviates. The soil texture, however, is the same under and inside the respective measuring instrument. Each column has a dimension of 1 m$^2$ surface area and 1.5 m depth. The weight of each column is measured with three precision shear stress load cells (Model 3510, Tedea-Huntleigh, Canoga Parl, CA, USA) with a measurement precision of 0.01 kg, like in ES-LMa*. The lysimeters were filled monolithically by the preparative method (Pütz and Groh, 2023b) preserving the natural soil structure and the lysimeter stations were installed in 2010. Pressure at the bottom of the lysimeter was generated by a bi-directional pumping mechanism that allowed either drainage into an external water reservoir (weighted tank) or inflow into the lysimeter from this reservoir, depending on the pressure difference between the lysimeter and the surrounding field soil at 1.4 m depth. Both the pressure head in the field and the bottom of the lysimeter were measured with a tensiometer (TS1, UMS, Munich, Germany). $SWC$ is measured within each lysimeter at a depth of 0.1 m below the surface with time domain reflectometry probes (CS610, Campbell Scientific, North Logan, UT, USA) at a resolution of 0.1 % SWC, according to the manufacturer. More details on the technical specifications of lysimeter facilities within the SOILCan are described by Pütz et al. (2016), on excavation methods in Pütz and Groh (2023b), and the Selhausen facility in (Groh et al., 2022).

Lysimeter raw weights underwent manual and automatic plausibility checks and periods with fieldwork/maintenance were removed. The lysimeter raw data were corrected for the pumping activities across the lower boundary system. To further reduce the impact of noise on the determination of the land surface water fluxes the Adaptable Window and Adaptable Threshold (AWAT) filter routine was applied at both sites. The AWAT filter handles non-stationary measurement errors in the lysimeter raw weight time series (Peters et al., 2014, 2016, 2017). In this three-step process, we employ adaptive techniques to smooth the time series by adjusting the width of the time window for the moving average. Moreover, adaptive threshold values are utilized, considering both the signal strength and noise levels. The evaluation of noise and signal strength is performed by analyzing a moving polynomial and subsequently examining the residuals for each data point. This enables us to accurately determine the presence of noise and the strength of the signal. In the third step, we identify local maxima and minima and incorporate them to prevent slight yet consistent underestimation during changes in the flux direction. This aspect is particularly crucial for the precise detection of minor flux events such as dew or SVA. The details of the AWAT filter are described in Peters et al. (2014, 2016, 2017), and its application on lysimeter raw data in Paulus et al. (2022) for ES-LMa*, and Schneider et al. (2021),

for DE-RuS.

Based on Paulus et al. (2022) the direction of the lysimeter weight change in each time step ($\Delta W$, mm time$^{-1}$), is used to classify them into one flux category, assuming that there is only one dominant flux during each time step (5 minute at ES-LMa* and 1 minute at DE-SeH) with:

$$\Delta W < 0 = \qquad\qquad evaporation$$

$$\Delta W > 0 = \begin{cases} rain & > 0\,mm \\ rH & > 95\,\% \\ T_{dew0.1m} & > T_s \\ T_{dew0.1m} & < T_s \end{cases} \qquad \begin{matrix} rain \\ fog \\ dew \\ SVA \end{matrix}$$

We calculated dewpoint temperature ($T_{dew}$, °C) from air temperature ($T_a$, °C) measured at a height of 1 m (Sonntag, 1990). Since the average vegetation height and hence the level where dew condensation occurs is at 0.1 m we estimated $T_{dew0.1m}$ = $T_{dew1.0m}$ - 1.4 °C. This calculation was based on a campaign-based comparison between the $T_a$ sensors at 1 m height and 0.1 m height above the surface (see Paulus et al. (2022) for further details) on soil or plant surfaces ($T_s$). For ES-LMa*, we additionally chose a last node with the category "residuals".

The lysimeter time series from Majadas de Tiétar comprises four years of data from 1 January 2018 to 31 December 2021. The time series from Selhausen comprises three years of data from 1 January 2018 to 31 December 2020. Please note again that on both sites, none of the lysimeters are below the EC stations (see Fig.2).

### 3.3 Auxilliary measurements

Additional hydro-meteorological measurements were analyzed at both sites at a temporal resolution of 30 min. At ES-LMa*, meteorological variables monitored were $T_a$ and $rH$ (capacitive humidity sensor CPK1-5, MELA Sensortechnik, Germany), both collected at 1 m height above surface level. $T_{dew}$ and atmospheric vapor pressure ($e_a$, hPa) were calculated based on $T_a$ and $rH$ (Sonntag, 1990). Precipitation was measured with a weighing rain gauge (TRwS 514 precipitation sensor, MPS systém, Slovakia) and mole fraction of water vapor in dry air ($\rho$, mmol mol$^{-1}$) were measured in a profile of 4 levels (0.1, 0.5, 1, 2 m) (LI-840 CO 2/H2O Analyzer, LI-COR Inc., Lincoln (Nebraska), USA).

Short- ($SW$, W m$^{-2}$) and longwave ($LW$, W m$^{-2}$) downwelling (SW$_{IN}$, LW$_{IN}$) and upwelling (SW$_{OUT}$, LW$_{OUT}$) radiation of the herbaceous layer was observed with a net radiometer (CNR4, Kipp and Zonen, Delft, The Netherlands) at a measurement height of $\sim 3$ m. $T_s$ is calculated from $LW$ and all equations used for the conversion of meteorological vari-

ables are given in Appendix B). $T_{soil}$ (PT-100, Jumo, Fulda, Germany), and $SWC$ (Delta-ML3, Delta-T Devices Ltd, Burwell Cambridge, UK) were measured outside the lysimeters at 0.05 m, 0.10 m, and 0.2 m depth at a resolution of 0.02 $^\circ$C $T_{soil}$ and 0.1 % $SWC$. Phenological shifts of the grass layer in ES-LMa* were examined based on green chromatic coordinates (GCC) from PhenoCam. For details regarding the camera setup and the computation of this specific vegetation index, we refer to the comprehensive description provided by Luo et al. (2018).

At DE-RuS, $T_a$ and $rH$ were measured at EC sensor height ($\sim 2.5$ m, HMP45C, Vaisala, Vantaa, Finland) and precipitation at 1 m with a weighing gauge (Pluvio2L, Ott, Kempten, Germany). $SW_{IN}$, $SW_{OUT}$, $LW_{IN}$, and $LW_{OUT}$ above the canopy was measured with a net radiometer (NR01, Kipp and Zonen, Delf, the Netherlands) at EC sensor height ($\sim 2.5$ m). $SWC$ was measured at 0.025 m depth at a resolution of 0.1 % SWC, according to the manufacturer (CS616, Campbell Scientific, Logan (Utah), USA). Conversions to other required variables were performed as described above for ES-LMa*.

### 3.4 Selection of time periods

Since we are particularly interested in the nighttime water fluxes, we compute diel aggregated values (e.g. mean or median conditions, summed flux) from noon to noon (instead of midnight to midnight). Consistent with the classification of fluxes of the lysimeters, we excluded days with rain, fog, and dew formation based on the following criteria: rain = 0; $rH < 95$ %; $T_{dew}$ 0.1 m $< T_s$. The final selection comprised 641 days in ES-LMa* and 98 days in DE-RuS. Previous observations of SVA in ecosystems occurred after the highest position of the sun, mostly at night. Therefore, we consider phases of different radiation conditions separately. We distinguish between the following periods

1. **day** when the sun is at an angle larger than 6° above horizon

2. **twilight** from golden hour (sun at 6° above horizon) to the end of astronomical twilight (sun at 18° below horizon)

3. **night** between the end of astronomical dusk and the beginning of astronomical dawn

4. **diel** from noon to noon

We used the function *getSunlighttimes* from the R software package suncalc (version 0.5.1 Thieurmel and Elmarhraoui, 2022) to determine the time of the day of the respective sun positions based on astronomical algorithms and the coordinates of the field site.

$\lambda E$ fluxes were quality checked according to Mauder and Foken (2011); Rebmann et al. (2005) and data with quality flag 0 and 1 were retained for further analysis. As opposed to $CO_2$ fluxes, $\lambda E$ fluxes are not regularly filtered for low friction velocity ($u^*$, m s$^{-1}$) conditions. However, to be conservative we removed the half-hours with the $u^*$ values below the critical $u^*$ threshold ($u^*_{thres}$, m s$^{-1}$) determined using the REddyProc package. To take into account the uncertainty introduced by the $u^*$ filtering, we repeated the analysis using the 5th and 95th percentile ($u^*_{thres,05}$ and $u^*_{thres,95}$) of the $u^*_{thres}$ estimate (Papale et al., 2006; Wutzler et al., 2018, $u^*_{thres}$ given in Appendix table F1).

For each lysimeter and half hour, the number of SVA observations was counted individually. If during the half-hour at least 20 minutes were classified as SVA, the half-hour was counted as SVA-dominated (individual column). Since days with dew,

fog and rain were filtered out, the remaining (non-SVA) 10 minutes can only contain evaporation measurements. Then, for
each half hour, we counted the number of lysimeters that detected SVA.

## 3.5 Comparing downward water fluxes detected with lysimeters and Eddy Covariance measurements

We will use $F$ (mm per unit of time) to represent water fluxes measured by the respective measurement method where flux
direction is indicated in the subscript. Thus, $F_{OUT,EC}$ and $F_{OUT,LYS}$ indicate evaporation, whereas $F_{IN,EC}$ are negative (i.e.,
downward-directed) $\lambda E$ fluxes and $F_{IN,LYS}$ are positive lysimeter weight changes, classified as SVA observations.

We investigated (i) the temporal consistency of the $F_{IN}$ between methods and (ii) the magnitude/comparability of the
measured $F_{IN}$ totals. To assess (i) temporal consistency, we count whether and how many weighing lysimeters detect $F_{IN,LYS}$
at the time of the occurrence of $F_{IN,EC}$. We compute precision and recall metrics (given in Appendix B). To examine the
concurrence among instruments concerning the seasonal onset of SVA-dominated nights, we identified the first period each
year during which five consecutive days exhibited more than four hours of $F_{IN}$. At the diel scale, we compared the timing of
the first and the last observation of $F_{IN}$ for each night. To compare (ii) the magnitude of the flux totals, we compare the half-
hourly mean absolute error (MAE, mm halfhour$^{-1}$) between the lysimeter median and the EC measured value (i.e., different
methods, different vertical and horizontal locations), as well as between the individual lysimeter columns (i.e., same method,
different horizontal locations).

$$MAE = \sum_{i=1}^{D} |F_{EC,i} - F_{LYS,i}| \tag{3.1}$$

Since the measurement location of the two methods is located at a vertical separation about two meters from each other,
a temporal shift and an attenuation of the signal are possible. Therefore, in addition to half-hourly measurements, we also
compare the diel sums between techniques for different subsets of the data: a) all night (quality filtered) $F$, b) all ($u^*$ and
quality filtered) $F_{IN,EC}$, and c) all ($u^*$ and quality filtered) $F_{IN,EC}$ during simultaneous $F_{IN,LYS}$ across all lysimeters. For
the comparison, we use Pearson correlation coefficient (R), MAE, coefficient of determination (r$^2$), and root mean square error
(RMSE, mm time$^{-1}$). Additionally, we compare the slope and the intercept using major axis regression ($F_{IN,EC} \sim F_{IN,LYS}$)
(MA) which was performed with the R-package lmodel2 to take into account that the uncertainty in the y and x axes are
comparable (Legendre, 2018).

Heterogeneous vegetation structures create micro-meteorological differences which in turn affect $F$. To assess whether
the differences between the EC and lysimeters ($\Delta_{LYS,EC}$) in ES-LMa* can be better explained by variations in micro-
meteorological factors or by variations in the soil hydraulic conditions we used a feature selection model with $\Delta$ LYS, EC
as the dependent variable (Jung and Zscheischler, 2013) and the predictors given in Appendix C. The list of given predictor
variables can be grouped into four distinct categories: meteorological conditions, the uncertainty of the EC technique, soil
conditions, and heterogeneity across lysimeters. Note, that the structure of the underlying data causes differences in the in-
formation content between the variable categories. Heterogeneity across lysimeters incorporates spatiotemporal information,

while all other categories only contain temporal information. Due to gaps in the lysimeter auxiliary measurements, the year 2018 was excluded from this part of the analysis.

The advantage of the feature selection method is that it is suitable to distinguish the importance of individual features although there is a high correlation within the set of given features, which is the case for many soil-hydro-meteorological features. Feature selection was performed using Random Forest (Breiman, 2001) as a first modeling step (100 trees) on a subset of predictor variables and using the out-of-bag estimate to calculate the cost function. Then, an ensemble of equally good models was selected (all models with mean squared error (MSE) > min(MSE) + 1 sd(MSE)) accounting for the performance differences based on the stochasticity of the Random Forest method. To explain the effects of individual predictors identified with the feature importance on $\Delta$ LYS, EC we used SHapley Additive exPlanations (SHAP) values (Lundberg and Lee, 2017). SHAP values were calculated on the unseen test data in a 10-fold cross-validation. We tested two model versions with *model.v1*: only providing spatiotemporal variables and *model.v2*: additionally providing lysimeter ID as a categorical input variable. Potential $SWC$-related thresholds in the diel relationship between SVA and evaporation, were assessed by employing piecewise linear regression. The threshold is defined as the breaking point between two linear models fitted separately to the data obtained from the EC and the lysimeter to test if these thresholds are consistent across the two methods.

## 4 Results and Discussion

### 4.1 Seasonal and diel meteorology

In the semiarid site ES-LMa*, the $F_{EC}$ fluxes follow a pronounced seasonal cycle (Fig. 3a, for acronyms see table A). The largest $F_{OUT,EC}$ fluxes occur every year between March and June. During this period, (i) soil water supply is high as soil moisture is replenished after winter and (ii) soil water demand is also high as sufficient energy is available for evaporation and vegetation is active (Fig. 3c,e). Each year around the end of May $SWC$ declines sharply in response to reduced precipitation (Fig. 3b,c). Consequently, evaporation is reduced, leading to lower $rH$ and consequently an increase in atmospheric demand. Within a couple of days, greenness decreases, indicating the withering of the grasses, while the diel amplitude of $T_s$ increases (Fig. 3a,d,e).

When $SWC$ is high, $F$ oscillates around zero between sunset and sunrise. In contrast, when soil is dry a night-time $F_{IN,EC}$ emerges shortly after the daytime evaporation declines (Fig. 3a). This pattern is most obvious in the second half of the night. This observation was confirmed by the lysimeter records across all four years of observations: night-time weight increases during this period occurred between sunset and sunrise and were classified as SVA (Fig. 3b). An illustration of the daily measurements from both instruments over four days of the dry season is shown in Appendix Fig. D1. It shows that $rH$ remains below 70 % and $T_s$ never reaches $T_{dew}$, which shows that the $F_{IN}$ is not related to fog deposition or dew formation.

The seasonal cycles of $F_{EC}$ in the temperate site DE-RuS are different from ES-LMa*. Here, the annual period of active daytime $F_{OUT,EC}$ lasts longer, e.g. from February until November (Fig. 4a). Strong changes in the $F_{OUT,EC}$ during summer are related to crop management (Fig. 4a,e) revealing substantial differences between the years 2019, compared to 2018 and 2020. While in 2019, the $F_{OUT,EC}$ is consistently high over the whole summer, in 2018 and 2020 it is sharply reduced in

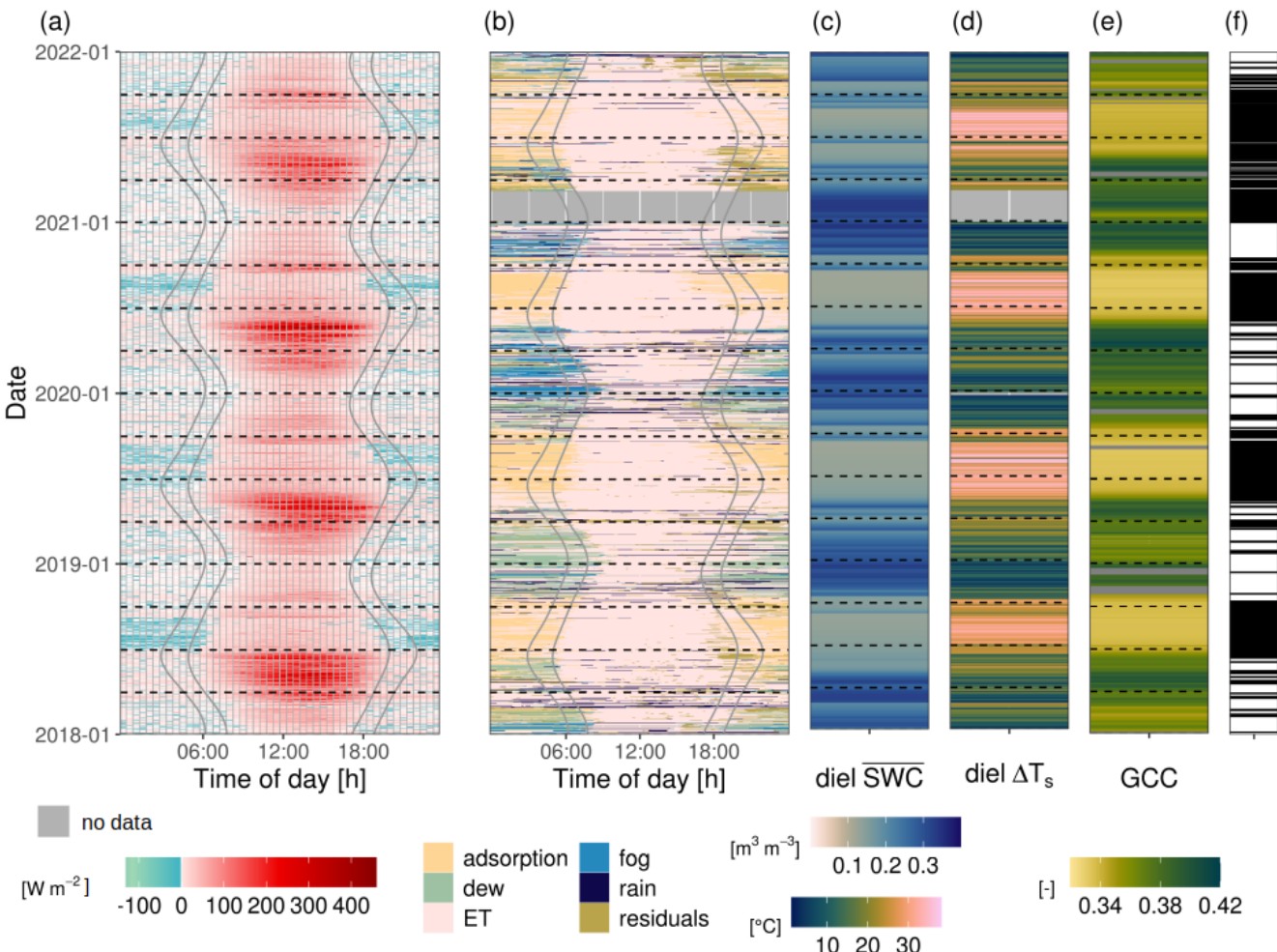

**Figure 3.** Diel and seasonal dynamics of (a) quality filtered latent heat fluxes from the eddy covariance method (b) dominant lysimeter fluxes (exemplary shown for L6) at the Majadas de Tiétar field site. Solid vertical lines mark the end of the night, sunrise, sunset, and beginning of the night, respectively (determined with the geographic coordinates of the field site). Panel (c) shows diel means of volumetric soil water content at 0.1 m depth (diel $\overline{SWC}$) and (d) maximum diel difference in surface temperature ($\Delta T_s$). Green chromatic coordinate values (GCC) for the grasses are shown in panel (e). In panel (f) the dates selected for this comparison based on the absence of rain, fog, and dew are marked as horizontal black lines (see section 3.4).

July associated with the harvest of the crops (Fig. 4a,e). Similarly to ES-LMa* this reduction is followed by several weeks of increased diel $T_s$ difference reaching values of more than 30 $^\circ$C between day and night during bare soil conditions with harvest-residuals in the EC source area (Fig. 4d). In contrast, in the summer of 2019 such extreme $T_s$ differences were only occurring on individual days, likely because the soil was wet enough near the surface to keep bare soil evaporation close to potential evaporation. The nighttime fluxes in DE-RuS oscillate around zero during wet conditions but as opposed to ES-LMa* this is also the case during dry conditions. The lysimeter records confirm that in DE-RuS, less frequent $F_{IN}$ during the night occurs compared to ES-LMa* in all seasons. Lysimeter weight increases are only sporadically during individual days and a short number of hours classified as SVA. The only exception is a period of two weeks in 2018 right after the harvest.

The different conditions in the two ecosystems and the fluxes associated with the lysimeter weight changes confirm that, while SVA is a frequent flux in ES-LMa* across years, it occurs only occasionally in the temperate agricultural ecosystem. The patterns in the EC observations also support these findings.

These results show that SVA in DE-RuS only occurred during a few weeks in the year 2018. In this time period (2018-07-20 until 2018-08-22) the Standardized Precipitation Evaporation Index (aggregated over 30 days; $SPEI\_30$) at DE-RuS indicates extreme drought (Appendix Fig.E1, (Svoboda et al., 2002)) (Pohl et al., 2023, 2022). Such dry conditions during annually more than two weeks have been recorded at this site only five times since 1950. However, out of these five three times occurred since 2010 (2011, 2018, 2020) (Pohl et al., 2022). The results from the temperate ecosystem confirm statements from classic literature that SVA is strongest in central European climate conditions in late summer when the soil is dry and uncovered (Blume et al., 2016b). At ES-LMa*, in contrast, SVA was observed each summer, but the years of investigation contained "only" moderate and severely dry periods (Appendix Fig.E1a) suggesting SVA to be the norm in the semi-arid area. This indicates that under the current climate SVA in temperate (agricultural) ecosystems only occurs during extremely dry conditions with no, or only little vegetation. It underscores that while the probability of occurrence of SVA is influenced by climate (i.e. more common in semi-arid and arid regions), it can also occur in more humid regions. This is because it depends on soil-intrinsic physical properties, such as texture (clay content, clay mineralogy, and organic carbon content) (Orchiston, 1954; Arthur et al., 2019; Yukselen-Aksoy and Kaya, 2010), soil structure that affects vapor transport characteristics (i.e. soil diffusion coefficient), and can happen anywhere if the dynamic requirements like temperature and moisture gradients are met. Considering the current climate change and increase in aridity foreseen in models, the importance of SVA might become more prominent also in temperate ecosystems.

The vertical gradient of $\rho$ between 0.1 and 2 m height above the soil during nights in ES-LMa* was investigated separately for conditions of $F_{OUT,EC}$, and $F_{IN,EC}$, relative to the diel mean $\rho$, respectively (Fig. 5). During the occurrence of $F_{OUT,EC}$, the air is relatively dry compared to the 24h mean, but wetter towards the soil surface. During the occurrence of $F_{IN,EC}$, it is the opposite situation, with the air at 2 m height being relatively moist but dry towards the soil surface. These measurements independently indicate that under conditions of $F_{OUT,EC}$, the air close to the soil is wetter than the atmosphere whereas under conditions of $F_{IN,EC}$ it is drier. From a gradient perspective, the latter case creates a vapor flux towards the soil, as described in the theoretical example in Section 2 Fig. 1. The measurements also indicate that $F_{IN,EC}$ are predominantly related to processes happening at the soil surface and to a lesser extent by the subsidence of dry air masses from the higher atmosphere,

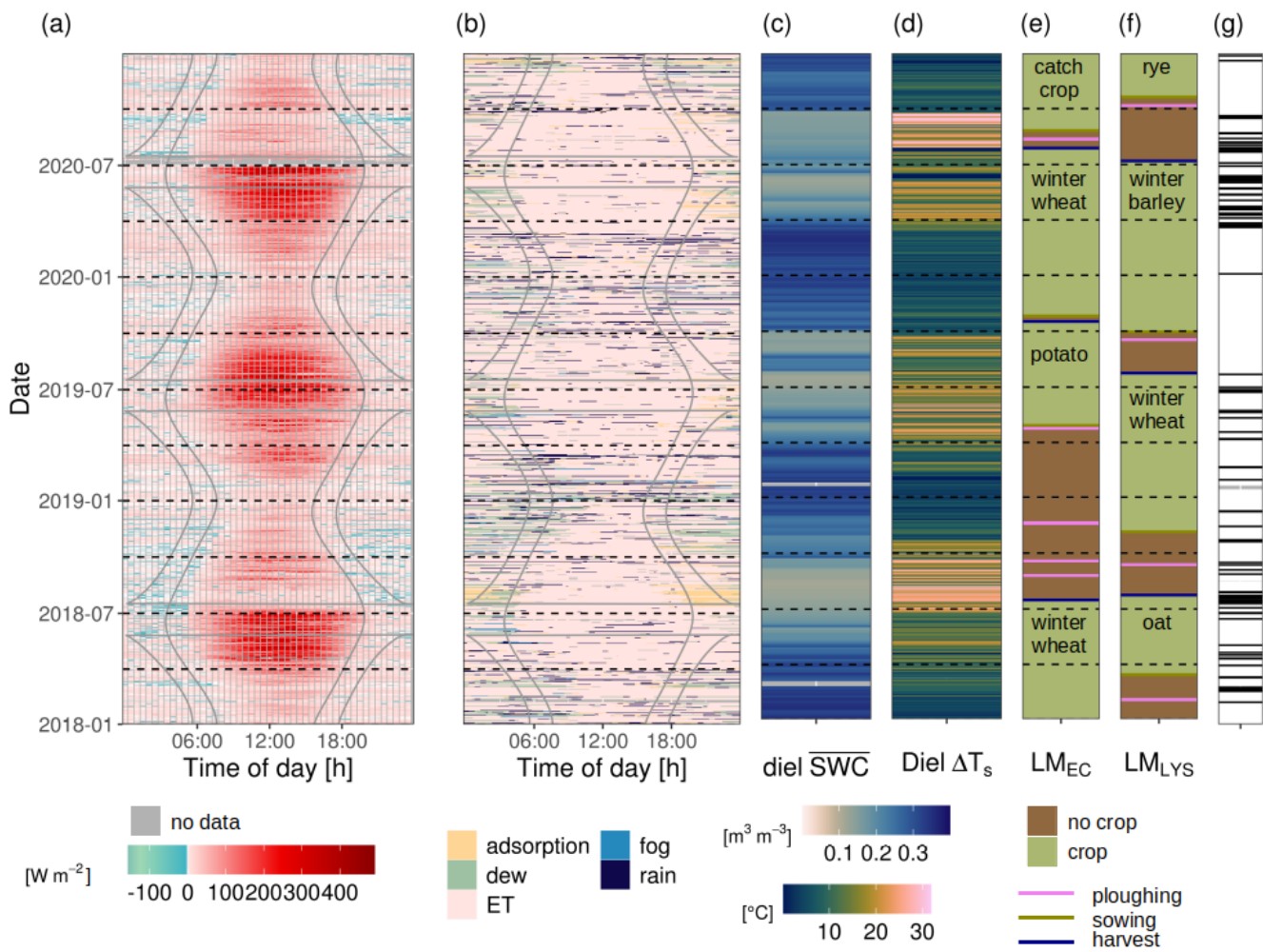

**Figure 4.** Diel and seasonal dynamics of (a) quality filtered latent heat fluxes from eddy covariance (EC) technique (b) classified dominant lysimeter (LYS) fluxes (exemplarily shown for $Se\_Y\_032$) at the Selhausen agricultural field site. Solid vertical curves mark the end of the night, sunrise, sunset, and beginning of the night, respectively (determined by the geographic coordinates of the field site). Mean volumetric soil water content at 0.1 m depth $\overline{SWC}$ and maximum diel difference in surface temperature ($\Delta T_s$) are displayed as diel measurements in panels (c) and (d). Land management (LM) is illustrated separately below the EC (e) and on the LYS (f). In panel (g) the dates selected for this comparison based on the absence of rain, fog, and dew are marked as horizontal black lines (see section 3.4).

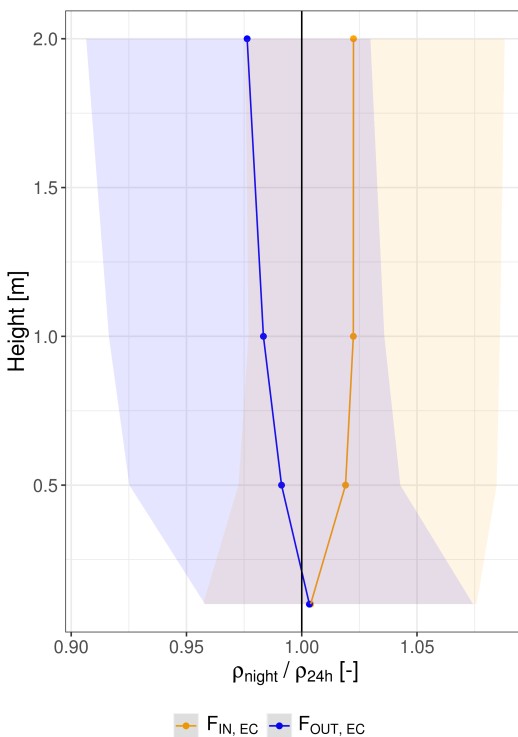

**Figure 5.** Vertical profile of mean nighttime absolute humidity ($\rho_{night}$) divided by the mean diel absolute humidity ($\rho_{24h}$) across all heights. The points and shaded areas illustrate the median and inter-quartile range during moments at night when $F_{OUT,EC}$ in blue and $F_{IN,EC}$ in yellow.

because the $\rho_{night}/\rho_{24h}$ profile between 1 and 2 m height is stable. Since the distinction between the micro-meteorological conditions shown in Fig. 5 is based only on observations of EC, this result (based on $\rho$ as an independent observation) supports our hypothesis ii) that EC can detect SVA.

## 4.2 Temporal patterns in the flux direction: consistency among instruments

We compared the flux directions measured with both instruments to investigate the consistency between measurement methods. The results are summarised in Table 2. At ES-LMa*, 4017 half-hours of $F_{IN,EC}$ were observed on 448 days, which is 30 % of the total measurement period. During 67 % of these EC observations, three or more (>50 %) of the lysimeters measured SVA. During 88.5 % of the measured $F_{IN,EC}$ fluxes simultaneously at least one of the lysimeters measured SVA. Applying the $u^*_{thres}$ value to filter out non-turbulent conditions removes 56 % of the half-hourly measurements. The agreement between measurement methods after $u^*$ filtering differs only marginally, which is consistent across different $u^*_{thres}$ (see Appendix Tab. F2). Excluding daytime and twilight increased the relative agreement with lysimeters by 6 %.

Between 89 % and 71 % (depending on the number of lysimeters considered) of all measured $F_{IN,EC}$ (before $u^*$ filtering) are in agreement with $F_{IN,LYS}$ as the reference ground truth (precision). Of all $F_{IN,LYS}$, however, only 53 % are detected by the EC instrument (recall). The recall rate increases to 75 % when all lysimeters are in agreement about the flux direction. These results suggest that in ES-LMa*, the great majority of $F_{IN,EC}$ are signals of SVA and that the EC method tends to underestimate the number of half-hours with $F_{IN}$ detected by lysimeters by at least 25 %. This could be partly related to a strong spatial heterogeneity of SVA, with EC performing best when SVA occurs across the field site, and not only in a few locations.

This can be related, on the one hand, to the high spatial variability of the soil conditions and, in some cases, the conditions at a lysimeter location are not representative of the whole ecosystem, and, on the other hand, to the variability of the source area of the fluxes measured by eddy covariance (eddy covariance footprint), whose shape and orientation depend on the wind direction and turbulence conditions, the first of which are highly dynamic in ES-LMa (see El-Madany et al., 2018). Although the lysimeters were placed in an area representative of the ES-LMa* footprint, they were not placed in the immediate vicinity of the EC instruments to avoid disturbances. Therefore, we expect a better agreement between lysimeters and EC data only when all lysimeters agree in SVA detection, and thus when the process occurs in many locations on the site and is therefore more likely to be detected by the EC independently of the shape and orientation of the footprint area.

**Table 1.** *Comparison of the number of simultaneous observations of flux direction towards/into the soil between EC and lysimeters for different filter criteria for ES-LMa\* and DE-RuS.*

| | | | LE < 0 + meteo | LE < 0 + meteo + u* | LE < 0 + meteo + u* + twilight + night | LE < 0 + meteo + u* + night | LE < 0 + meteo + u* + no crop | LE < 0 + meteo + u* + extreme drought |
|---|---|---|---|---|---|---|---|---|
| ES-LMa* | n night | | 448 | 380 | 375 | 318 | n/a | n/a |
| | n halfhours | | 4017 | 1752 | 1664 | 1066 | n/a | n/a |
| | | 0 | 461 (11.5%) | 225 (12.8%) | 193 (11.6 %) | 121 (11.4%) | n/a | n/a |
| | n SVA halfhours | 3 | 2676 (66.6%) | 1192 (68.0%) | 1166 (70.1%) | 802 (75.2%) | n/a | n/a |
| | | 5 | 1115 (28.8%) | 829 (26.9%) | 466 (27.5%) | 338 (31.8%) | n/a | n/a |
| DE-RuS | n night | | 165 | 31 | 29 | n/a | 16 | 10 |
| | n halfhours | | 239 | 93 | 75 | n/a | 49 | 39 |
| | | 0 | 151 (63.2%) | 57 (61.3%) | 43 (57.3 %) | n/a | 22 (44.9%) | 15 (38.5%) |
| | n SVA halfhours | 3 | 33 (13.81%) | 16 (17.2%) | 16 (21.3 %) | n/a | 16 (32.7%) | 14 (35.9%) |

In DE-RuS 239 half-hours of $F_{IN,EC}$ were observed on 165 days, which is 15 % of the measurement period. In contrast
to ES-LMa*, for 63 % of the $F_{IN,EC}$ half-hours, no SVA was detected by the lysimeters. Filtering with the $u^*_{thres}$ and for
phases of twilight or night slightly decreased the number of hours matching lysimeter SVA, this could however also be an effect
of the reduction of the sample size, which amounts to 40 % after $u^*$ filtering. The agreement between methods increased under
conditions of bare soil and extreme drought despite a strong reduction in the sample size to only 16 and 10 days, respectively.
The highest agreement was found for conditions of extreme drought but even then, 39 % of the $F_{IN,EC}$ were not accompanied
by lysimeter SVA. Under such conditions, only 39 half-hours from 10 days were available for comparison.

One potential reason for this difference between the sites is different crop and crop residue management in DE-RuS since
the height of the vegetation influences gas exchange. SVA was reported to be reduced below or in the vicinity of tall, active
vegetation by 76 % (Kosmas et al., 2001). Also the larger distance between the instruments in De-RuS (357 m), as compared
to ES-LMa*, could have an effect on the results. Another reason could be that the topsoil in DE-RuS remains relatively wet as
compared to ES-LMA*, with a mean and standard deviation of SWC amounting to $16.8 \pm 6.6$ % and $7.8 \pm 4.8$ %, respectively.
DE-RuS remained much wetter than the semi-arid site even under extreme drought ($13.1 \pm 2.2$ %). At the same SWC under
controlled conditions, the soil from DE-RuS should theoretically have a similar or higher capacity to adsorb water than the soil
in ES-LMa* due to its high clay content (17 % compared to 5 %) which influences the water sorption behaviour more strongly
than the mineralogy for mixed soils with low kaolinite content (Arthur et al., 2015). Hence, these effects of the soil properties
don't come into play when the overall climatic conditions are too wet.

The results support our hypothesis i) that the EC method is able to capture the difference between the two sites in different
climates, detecting much less half-hours of $F_{IN,EC}$ at the temperate site. Since more data is available for the statistical compar-
ison of $F_{IN}$ between methods from ES-LMa*, compared to DE-RuS, we will predominantly concentrate on the methodological
comparison based on data from ES-LMa*.

The timing of the first observation of $F_{IN,EC}$ at the diel scale is consistent between years in ES-LMa*. Usually, $F$ turns
negative within the hour around sunset or later during the night (Fig. 3a and Fig. G1a). The last observation of $F_{IN,EC}$ is
usually around sunrise (Fig. G1b). However, there is a stronger delay observable in the morning, indicating $F_{IN,EC}$ often
continue within the first hour after sunrise. An explanation for this observation could be the shallow angle of the sun right after
sunset, delaying surface heating until it reaches a higher position in the sky. At the seasonal scale, we compared the agreement
between methods by defining the onset of prolonged $F_{IN}$ as more than 4 hours during at least 5 consecutive days. In ES-LMa*,
the lysimeters consistently detect this onset earlier during the years, compared to EC (Appendix Fig. G22). In 2018 and 2019,
the time difference was less than two weeks (13 and 9 days, respectively). But in 2020 it amounts to one month, and in 2021
nearly two months (32 days and 58 days, respectively). Since in 2020, the EC also detects prolonged $F_{IN,EC}$ earlier during
the year already, however only over the span of 3 consecutive days, this highlights that it strongly depends on the definition of
the onset. Nevertheless, when considering the prospective benefits of these outcomes, we believe that a definition that ensures
a more cautious assessment, as opposed to an overestimation, is preferable. A potential explanation for the mismatch between
methods in these two years is frequent rain events during the dry-down phase in 2020 and 2021, as compared to 2018 and 2019,

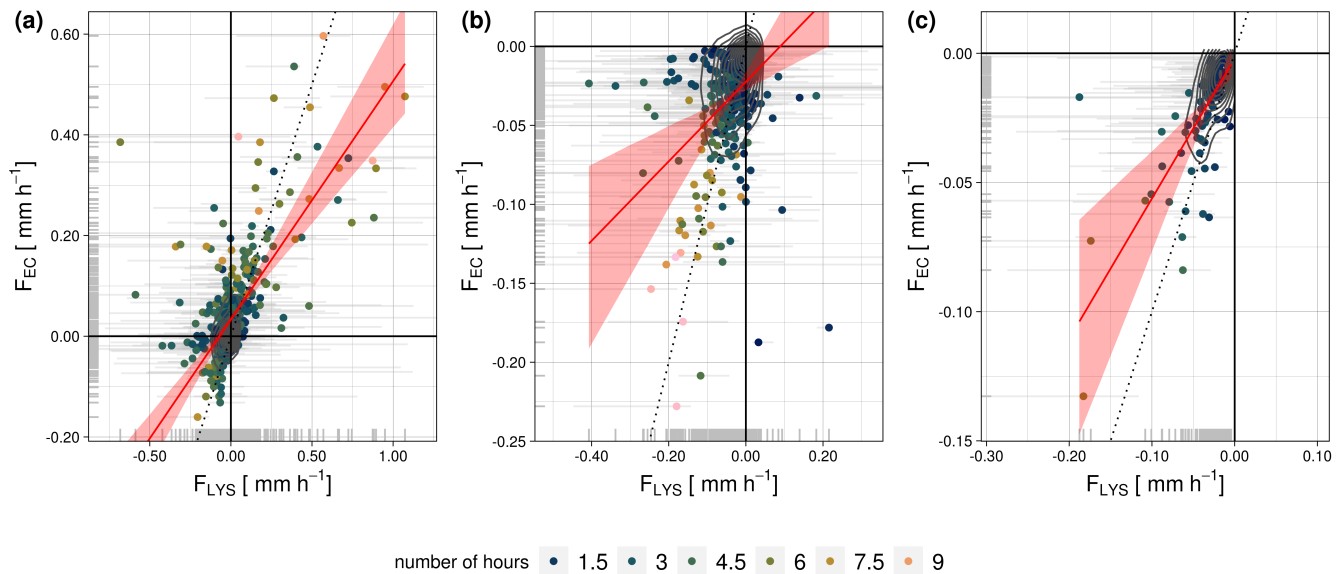

**Figure 6.** Comparison between night-time sums of lysimeter measured water fluxes ($F_{LYS}$) against eddy-covariance measured fluxes ($F_{EC}$) in Majadas de Tiétar (ES-LMa*, toprow) for different subsets of the data: (a) all good quality nighttime fluxes, (b) negative EC nighttime fluxes, and (c) negative nighttime fluxes and all lysimeter fluxes classified as soil adsorption of atmospheric water vapor. The red line illustrates a major axis regression model and the red shading the confidence interval of the model. The black dotted line illustrates identity. Horizontal grey lines illustrate the minimum and maximum sum observed from single lysimeter columns. The colorcode illustrates the number of hours over which this sum was formed. It depends on how many observations were measured for the respective conditions on each night.

**Table 2.** Statistics for the comparison of $F_{IN,EC}$ and $F_{IN,LYS}$ as nighttime sums in ES-LMa* with different filtering periods. *See also Fig. 6*

| Site | Filter | n | R | RMSE | MAE | intercept | slope | $r^2$ |
|------|--------|---|---|------|-----|-----------|-------|-------|
| | | | | | | [mm/ night] | | |
| ES-LMa* | night | 518 | 0.656 | 0.125 | 0.068 | 0.034 *** | 0.473 *** | 0.431 |
| | night + $F_{IN,EC}$ | 380 | 0.320 | 0.061 | 0.036 | -0.022 *** | 0.252 *** | 0.102 |
| | night + $F_{IN}$ | 108 | 0.706 | 0.027 | 0.017 | -0.002 *** | 0.543 *** | 0.489 |

affecting the flux amount to be below the limit of detection of the EC method, but not the lysimeter, as will be demonstrated in the next section.

## 4.3 Amounts of soil water vapor adsorption quantified by eddy covariance and lysimeter measurements

The comparison between the integrated nighttime $F$ sums is illustrated in Figure 6 and the respective statistical summary is given in Table 2. In ES-LMa* we find that $r^2$ and slope which describe the relationship between the mean lysimeter-measured flux magnitudes and the EC-measured flux magnitudes are similar for the case where all good quality $u^*$-filtered nighttime measurements are compared, including $F_{OUT}$ and $F_{IN}$ (Fig. 6 a), or only $F_{IN}$ (Fig. 6 c) are compared (0.431 and 0.495; and 0.473 and 0.543, respectively). This indicates that generally there is a strong dampening in the signal recorded by the EC method compared to the lysimeters but no systematic bias of the good-quality nighttime $F_{IN,EC}$, compared to the nighttime $F_{OUT,EC}$.

The strong dampening of the signal is only observed in ES-LMa*. In DE-RuS, there is generally a better agreement between lysimeter and EC fluxes, expressed by a strong correlation (0.858) when all good quality nighttime fluxes are considered (Appendix Fig. H1 and Tab. H1). However, the limitation in observation data (n = 6) does not enable us to draw any conclusions about the consistency of the pattern in DE-RuS when considering $F_{IN}$ only.

The EC method consistently underestimates $F$ at ES-LMa* compared to the lysimeters but there is also a great variation between individual lysimeters (grey bars in Fig. 6 and Fig. I1). Lysimeter L3, L5, and L6, and the EC method seem to have a much better linear relationship compared to lysimeter L1 and lysimeter L2, indicated by the scatterplot showing a straight line, close to the identity line. However, we find higher agreement between EC and the median across the lysimeters (Table 2) than between EC and individual lysimeters (Table I1). One interpretation of this result could be that each lysimeter covers a smaller spatial scale (1 m$^2$ each) compared to the EC (illustrated in Fig.2 as footprint climatology) but the average across lysimeters better represents the spatial mean and is therefore more in line with the EC observations.

Nevertheless, a systematic difference between the measuring instruments in the form of a bias remains. We evaluated the difference for different $u^*_{thres}$ (see Appendix Tab. F3). When considering only measurements above $u^*_{thres}, 95$ the strength of the correlation increases to 0.79 and bias decreases to 0.028 compared to the median $u^*_{thres}$. Choosing a low $u^*_{thres}$ of 0.01 m s$^{-1}$ increases the mismatch compared to the median $u^*_{thres}$. This is not surprising as under stable nighttime conditions the ratio between vertical and non-vertical (drainage and advection) movement of $F$ is expected to be smaller. As a result, a larger proportion of the total $F$ leaves the source area undetected by the EC sensor than in daytime measurements with good atmospheric mixing (Wohlfahrt et al., 2005). Therefore, as in the case of $CO_2$ fluxes, we can expect an underestimation of $\lambda E$ fluxes under low $u^*$ conditions, leading to the observed systematic differences, which are partially relieved when a more conservative (higher quantile of the $u^*_{thres}$ distribution) is used.

It is important to note that our results are based on negative $\lambda E$ observations only. Considering the low fluxes at night and the random uncertainty of the EC data (Hollinger and Richardson, 2005; Lasslop et al., 2008), we could bias the fluxes by removing values close to zero or slightly positive. We would like to disprove the hypothesis that the relationship between the lysimeter and EC observations is based only on the bias introduced by the random error in the EC with three details from our results: 1. all integrated flux sums (except one, on 12.08.2020) are more negative than the error propagation of the random error associated to each half-hourly EC measurement (illustrated in Fig. J1). 2. If the $F_{IN,EC}$ was mainly the sum of the negative

fraction of the random noise, it shouldn't be linearly related to $F_{IN,LYS}$ when the sum is calculated over the same length of hours. We find, however, that the linear relationship between $F_{IN,EC}$ and $F_{IN,LYS}$ is weak when considering only short time periods (i.e. one hour R = 0.05) and strong when considering longer time periods (i.e. four hours R = 0.6). This indicates that for continuous measurements of $F_{IN,EC}$ a substantial part cannot be (solely) explained by noise. 3. The consistent strength in the statistical measures - irrespective of comparing all nighttime $F$, or only nighttime $F_{IN}$ (when we assume as a community that good quality nighttime $F_{OUT,EC}$ are valid observations, as is already the base of published work i.e. of Padrón et al. (2020) or Han et al. (2021)).

Although in this study we are dominantly interested in the differences in $F_{IN}$, the drivers of the fluxes and causes of the mismatch are the same as for $F_{OUT}$. Generally, the flux loss of EC has been acknowledged numerous times (Massman and Lee, 2002), often expressed in a non-closure of the energy balance (Foken, 2008; Mauder et al., 2020) and in a smaller magnitude measured by EC as compared to lysimeters. In a former study in ES-LMa* $F_{OUT,EC}$ amounted 35 % less compared to $F_{OUT,LYS}$ (Perez-Priego et al., 2017). This finding was independent of the spectral correction method for the EC (i.e. analytical (Moncrieff et al., 1997) or *in situ* (Fratini et al., 2012)). They suggested that the mismatch in dry periods in ES-LMa* could potentially be explained by strong radiation gradients due to the shade casted by the trees causing flux divergences. At a temperate site in the pre-alps, the underestimation of lysimeter evaporation with EC was 30 % (Mauder et al., 2018). Florentin and Agam (2017) reported from an arid desert with homogeneous surface conditions that nearly 50 % of the lysimeter fluxes were detected with EC for both, $F_{OUT}$ and $F_{IN}$. Although a definitive explanation couldn't be reached for the arid site, at the temperate site, the dissimilarity between the instruments was primarily attributed to the absence of energy balance closure in the EC system. Since there is a large variation in agreement between individual lysimeter stations in ES-LMa* we investigated the amount and potential drivers of the mismatch in the following section (Section 4.4).

### 4.4 Identification of the variables influencing the difference between lysimeters and EC SVA measurements

Figure 7a illustrates the distributions of half-hourly values of $F_{IN}$ for each individual lysimeter column and the EC instrument in ES-LMa*. The median of EC observations is lower than the median across all observations from individual lysimeters (-0.004 mm per hour; median-Lys). However, there is a large range in the observations also across individual lysimeters, revealing that the MAE between lysimeters is larger than between the two measurement techniques (Fig. 7b). A larger mismatch exists between EC and observations from station 1 (1L and 2L) compared to the other two stations. We investigated the potential reasons for the mismatch between the two instruments by means of a predictor variable selection procedure based on a random forest model analysis with the deviation between EC and lysimeter as the dependent variable (Jung and Zscheischler, 2013). Fig. 8a shows an estimate of variable importance based on how often each predictor variable was selected in the best models for model.v1. The four most frequently chosen variables were lysimeter $SWC$, $e_a$, $T_s$, and $\Psi_m$. Out of the 16 selected variables, 7 are related to soil temporal and 6 to soil spatiotemporal variability (soil variables measured in the lysimeter columns). These two groups of variables have also an overall stronger impact on the prediction (Fig. 8b) as compared to variables related to the temporal variability of atmospheric state or related to the uncertainty of the EC technique.

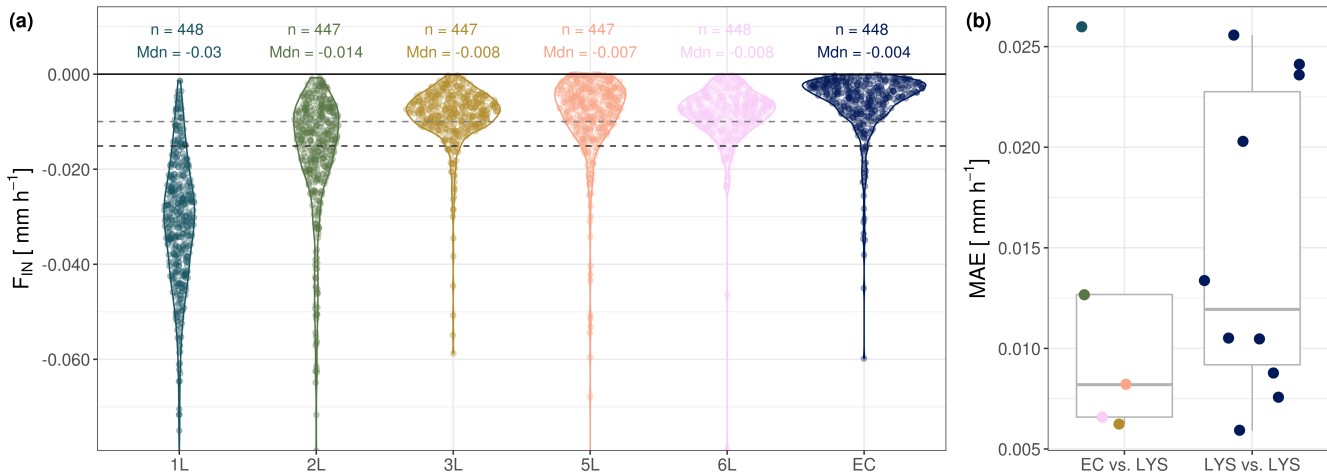

**Figure 7.** a) Distributions of half hourly readings shown individually for each lysimeter and the EC in Majadas de Tiétar. Only periods during which adsorption and negative latent heat flux were measured uniformly were selected. The horizontal dashed lines show the mean (black) and median (grey). b) Mean average error (MAE, mm) between individual lysimeter columns and EC (between techniques) and MAE between lysimeter columns (same technique).

The primary factor influencing the variation between instruments is $SWC$ within the lysimeters. The deviation between instruments decreases at lower $SWC$ (Fig. 8c) and higher $T_s$ (Fig. 8d). The fraction of variance explained of the random forest model is 0.449 according to the OOB score. In our analysis, this value is acceptable since we use it in an explanatory context and not for prediction, knowing that part of the variation between the two instruments is random noise. Interestingly, the model performance also does not substantially improve when lysimeter ID is provided as an input variable (*model.v2*), supporting the relevance of the $SWC$ within columns as main explanatory variable ($r^2$ = 0.449 and 0.438, rmse = 0.009 and 0.009 mm hour$^{-1}$, MAE = 0.004 and 0.004 mm hour$^{-1}$). Although lysimeter ID gets selected as a static predictor variable (see Appendix Fig. K1) the dynamics of soil moisture and temperature within lysimeters are more important to explain the observed difference between lysimeter and EC. This means that the lumped effect of static properties which might deviate between lysimeters such as clay or organic matter carries a lower information content for the prediction of the differences between instruments. Based on these results, it can be inferred that approximately 45 % of the discrepancy in $F_{IN}$ between the lysimeter and EC in ES-LMa* is dominantly influenced by the spatiotemporal variability of soil moisture and temporal variability of surface temperature.

Our finding that $SWC$ and $T_s$ are ranked as most important variables (based on their mean SHAP value) to explain the deviation between instruments is in line with SVA theory and other field observations. $SWC$ and $T_s$ are both drivers of SVA, controlling the strength of water retention as well as the vapor flux velocity. Several experimental studies confirmed small-scale variation in adsorption quantities of up to 100 % within a 4 m distance only due to soil exposure and the influence of the vegetation canopy (Verhoef et al., 2006; Kidron and Starinsky, 2019) and numerical models show that under dry conditions, diel temperature oscillations are substantial drivers of SVA (Saaltink et al., 2020). Here Fig. K1b and Fig. K1c show that the

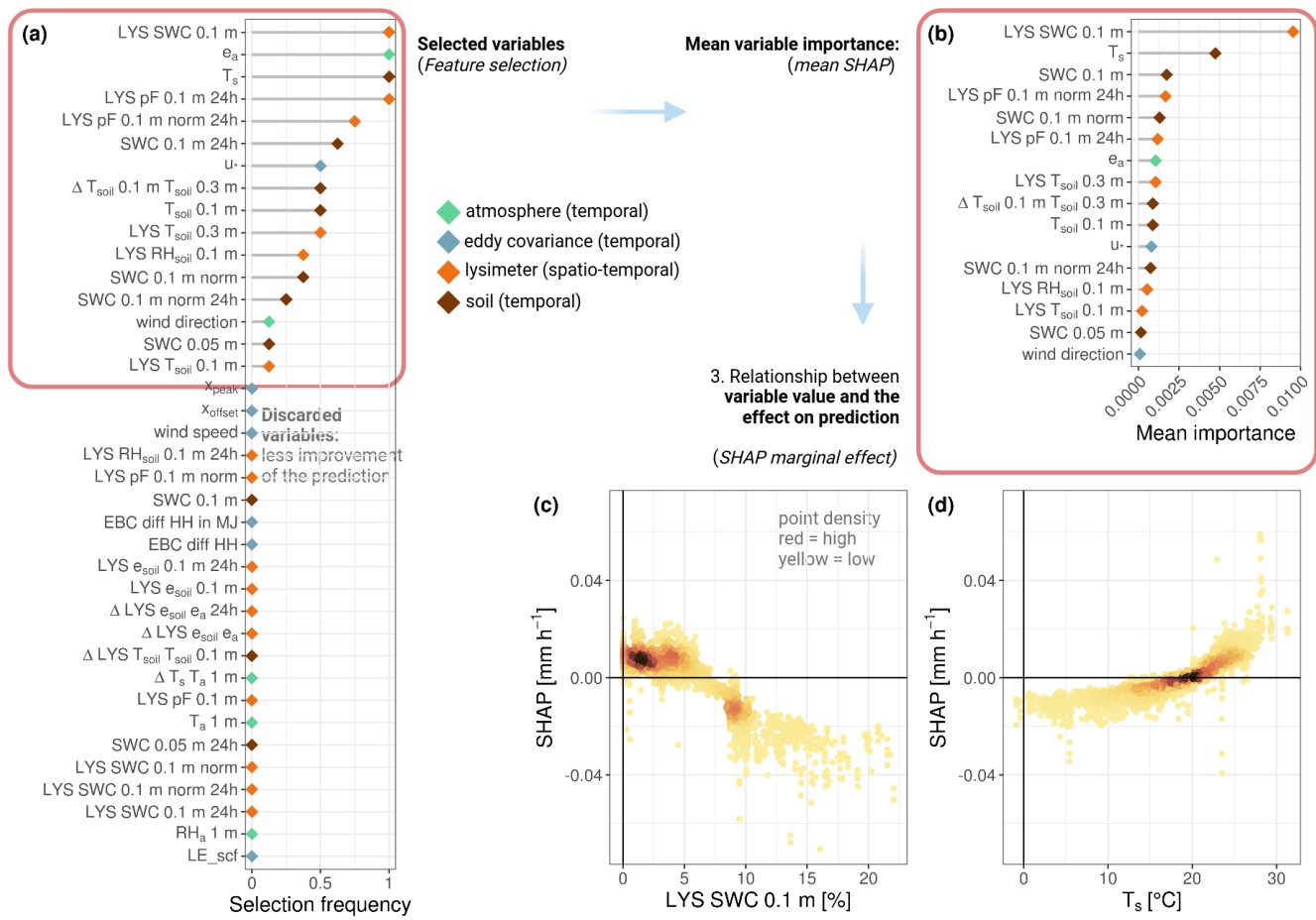

**Figure 8.** Panel a) depicts the selection frequency of the predictor variables of the best models from the first round of the feature selection procedure. The selected variables (indicated by the red rectangle) where subsequently incorporated in a model ensemble and their mean importance on the prediction is presented in panel b). Panel c) and d) display the marginal effects of the two most influential predictors, respectively. The full form and explanation of all variables is given in Table C1

.

$F_{IN}$ amount increases with lower lysimeter $SWC$ and higher $T_s$ and under these moments, the discrepancy between the instruments is reduced. One explanation for this effect could be a larger signal-to-noise ratio. Another explanation might be a higher spatial variability in $SWC$ for medium, than for dry conditions (Vereecken et al., 2007). Since Spanish tree-grass ecosystems (*Dehesas*) have a Savanna-like structure they are known to have very heterogeneous and patchy surface conditions due to the heterogeneous vegetation cover and fertility islands below and around the tree canopies that have very different

conditions in terms of soil properties compared to the open grasslands, which propagates into the surface energy and water balance. It is therefore possible that soil heterogeneity conceals the effect of variables associated with EC uncertainty on the

mismatch, which should be checked in a more homogeneous ecosystem. This is supported by the detectable effect of the $u^*$ that shows that the discrepancy between instruments decreases with higher $u^*$ (see Fig. L1), but its effect on the mismatch is one order of magnitude smaller than the effect of lysimeter $SWC$ and $T_s$.

Note that variables measured within the lysimeters carry additional spatial information content compared to the other variables, and hence their importance might be inflated. However, this is not the case for the soil-related variables, which still contribute substantially more compared to the EC uncertainty-related variables, suggesting that our conclusion that soil-related variables are more important than EC uncertainty-related variables is robust. It is further possible, that the spatio-temporal differences in soil hydraulic conditions of the lysimeters are caused by small differences in soil properties such as clay or soil

organic carbon content. Both variables are known to substantially increase soil sorption capacity and to generally affect soil water retention characteristics (Arthur et al., 2015, 2016). At the Majadas field site, the topsoil clay content is relatively constant between 0 and 5 % but an individual topsoil sample from outside the lysimeters contained 18 % clay. Such outliers in the spatial distribution of clay content have substantial non-linear effects on small-scale variations soil water retention characteristics at the dry end of the water retention curve and thereby could cause the observed variability of $SWC$ between lysimeters.

These results only reflect potential drivers of the differences between the two instruments during the times when SVA occurs, meaning that the model only receives input data from a very specific, filtered period of time. The drivers of the differences in $F_{OUT}$ are (potentially) different but are outside the scope of this analysis. Additional reasons for mismatch can be related to advection, non-closure of the energy balance, changes in the source area (extension and position of the flux footprint), or island effects of the lysimeters.

**4.5    Implications of soil water vapor adsorption for the soil water balance**

In the previous sections, we have demonstrated that $F_{IN,EC}$ under the selected conditions at our semi-arid site ES-LMa* carry a meaningful signal of SVA. In the last section of this manuscript we would like to build on these results and use the new opportunity to i) investigate the onset of SVA in ES-LMa* over a longer period of time with EC only and ii) investigate the importance of SVA for the diel soil water balance.

We investigate the onset of prolonged SVA determined based on EC observations in ES-LMa* for each dry season between 2015 and 2022 based on the hours per day of $F_{IN,EC}$ in Figure 9. The long-term data reveals the onset varying in time between 22. June, (2019), and 01. August (2020). However, it shows that there is a great interannual consistency in the $SWC$ decreasing to 0.1 when the period of $F_{IN,EC}$ starts (Fig. 9b). Further it shows that the onset always marks the end of the decrease of the evaporation flow (Fig. 9c).

These findings suggest that the dynamics we see in the EC observations correctly capture what is expected from the relationship between evaporation and SVA in the absence of transpiration, namely the onset of (prolonged) SVA coinciding with what Or et al. (2013) defined as the vapor diffusion-controlled Stage II evaporation. According to this concept, there is a so-called Stage I evaporation period, where the soil is wet and evaporation is dominantly limited or controlled by the atmospheric forcings (radiation, free flow, $rH$, and temperature). Usually, this phase is followed by a gradual decrease in evaporation (falling

rate period) when the soil surface has dried reflecting a transition to diffusion-limited vapor transport, with the dynamics of the

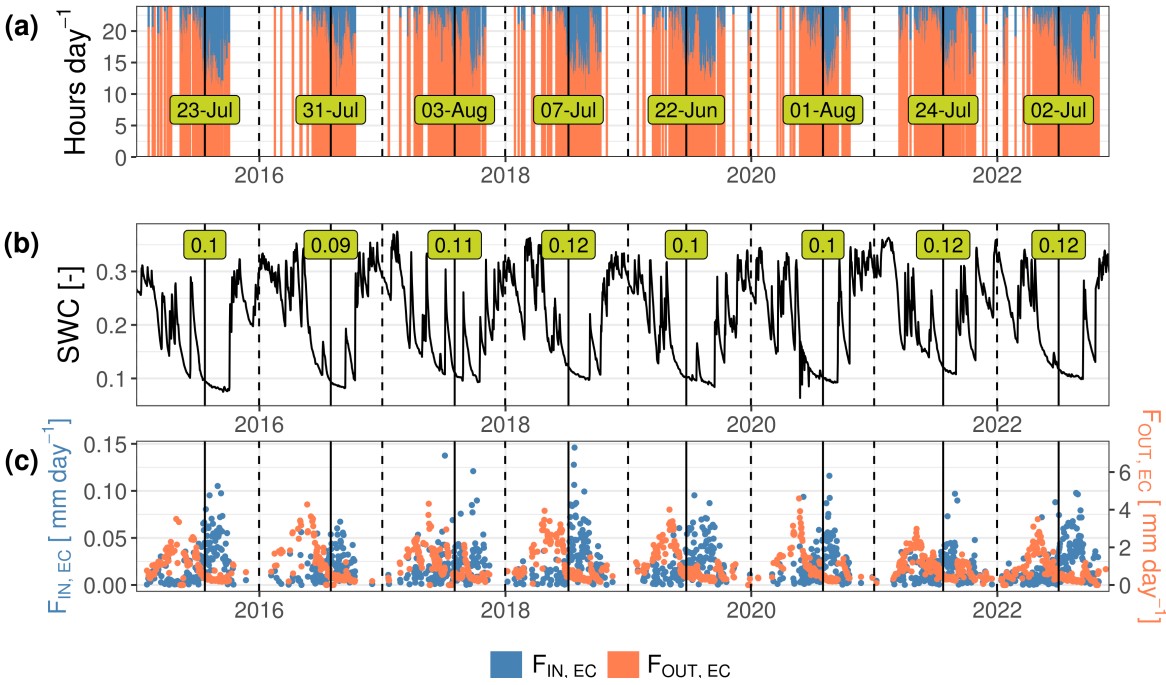

**Figure 9.** Panel a) illustrates the diel fraction of positive (red, $F_{OUT,EC}$) and negative (blue, $F_{IN,EC}$) $\lambda E$ fluxes measured with EC. The dashed vertical lines mark the onset of adsorption-dominated nights in ES-LMa*, defined as the first periods each year, where five consecutive days with more than four hours each of $F_{IN}$ were observed. The annotation in (a) gives the respective day for each year, with the respective soil water content (SWC) at 0.05 m depth given in panel (b). In panel (c) the evolution of the diel $F_{IN,EC}$ and $F_{OUT,EC}$, are presented as weekly means. In all panels, the solid vertical curves illustrate the threshold and the dashed vertical lines illustrate the beginning of the next year.

evaporation fluxes becoming stronger defined by the hydraulic properties of the porous medium (Or et al., 2013; Vanderborght et al., 2017).

Following this concept, this means that $F_{IN,EC}$ could help to identify the onset of film-flow dominated evaporation regime in the field. This is relevant information from a soil-physical perspective to correctly predict evaporation. It is also meaningful
from an eco-hydrological perspective since the disruption of the water-filled pore network in the topsoil and the decrease in $rH$ within the soil pores affects the soil biosphere i.e. when roots lose connection to water-filled pores (Passioura, 1988) or bacterial growth gets limited (Or et al., 2007).

Because $F_{OUT}$ decreases, and $F_{IN}$ increases over the dry period, the ratio of diel $F_{OUT}$ to diel $F_{IN}$ during Stage II increases with decreasing $SWC$ (Fig. 9 and Appendix M1). Figure M1 indicates that under Stage II evaporation, a substantial amount
of the diel evaporation in ES-LMa* might be composed of water that adsorbed during the night at the soil surface. At a $SWC$ below 7.8 % (estimated with piecewise linear regression) the EC method suggests the mean diel ratio to amount 0.09 with the

$95^{th}$ quantile amounting 0.25. This $SWC$ threshold is consistent with the lysimeter method (SWC, 7.0 %) but the lysimeters even record ratios of 0.27 and 0.64 (mean, $95^{th}$ quantile).

However, although it is obvious that the EC method underestimates both, (nighttime) evaporation and SVA, it should be mentioned that large weighing lysimeters could also overestimate both fluxes. Since the boundary conditions of the lysimeter are controlled at the bottom, the energy and water budget at the lysimeter surface might deviate from the surrounding soil (Kidron and Kronenfeld, 2017). More efficient heat loss of the lysimeter surface via nocturnal long-wave radiative cooling in the dry period would result in higher SVA. The extent to which heat loss through the walls of large weighing lysimeters affects SVA measurements still needs to be investigated (Paulus et al., 2022). Additionally, lysimeter fluxes are only a lumped information of mass changes caused by water fluxes, presumably at the upper boundary of the lysimeter, but temporal shifts in evaporation and condensation planes within the lysimeter (including the vegetation canopy) cannot be accounted for. Ultimately, lysimeter column-internal processes add to the uncertainty of what we use as "ground truth" in this study and need to be modeled, accounting for temperature and moisture gradients combined, to understand these processes. The most commonly used soil water retention curve models, relating $\Psi_m$ with $SWC$, i.e. the van Genuchten model, however, strongly underestimates the diel oscillations of $\Psi_m$ observed under natural conditions since it assumes a constant saturation in the dry end. As a consequence, the turbulent inward vapor flux into the soil and the modeled amount of SVA is heavily underestimated (Saaltink et al., 2020). Hence, soil water retention curves suitable to adequately represent the dry end are crucial when investigating how lysimeter internal evaporation-condensation processes might affect their measurements at dry conditions.

## 5  Conclusions

In this analysis we evaluated the possibility of detecting soil adsorption of atmospheric water vapor (SVA) using negative latent heat ($\lambda E$) fluxes from the eddy covariance method (EC) and evaluated it against lysimeters. We filtered EC measurements for periods without rain, fog, and dew in a Mediterranean and a temperate ecosystem. Using observations from large weighable lysimeters we could show that negative $\lambda E$ fluxes during conditions of low soil water content ($SWC$) contain signals of SVA in a Mediterranean tree-grass ecosystem, returning annually during the dry summer months. In this ecosystem, negative $\lambda E$ fluxes predominantly occurred during the night until the first hour after sunrise. We observed 448 nights with 4017 half hours of negative $\lambda E$ fluxes of which 88.1 % coincided with at least one lysimeter measuring SVA. Our results confirm that SVA at temperate sites is not as relevant and can only be observed under conditions of extreme droughts and the EC method was able to reproduce the differences between the sites. However, it detected substantially more often negative $\lambda E$ fluxes without lysimeters recording SVA, which might be related either to the larger distance and difference in managing practice between the instruments at the temperate site or an overall higher $SWC$ and smaller fluxes.

When lumped as nighttime sum, the difference in magnitudes of SVA measured with the lysimeter method and the EC method was the same as for nighttime positive evaporation fluxes. This is most likely related to the low aerodynamic turbulence during the night, where EC strongly underestimates the vertical flux. For friction velocity conditions, the strength of the correlation between methods increased and the bias decreased. At a half-hourly time scale, the spatial heterogeneity in SVA

magnitude measured among lysimiters was higher than among methods. This imposes limitations on the conclusions that can be derived from our experimental measurements in assessing the comparability of flux magnitudes. Nevertheless, since at the Mediterranean site the spatial pattern (amount of evaporation and SVA) is consistent, we assume the median fluxes across lysimeters reflect the spatiotemporal heterogeneity of the site.

This finding highlights a new measurement application of the EC method, namely that i) EC is able to capture the signal
of SVA, ii) EC tends to underestimate the occurrence frequency and the flux magnitude, and iii) the ability of EC to capture SVA likely is limited to ecosystems where $SWC$ decreases substantially below a threshold which in this study amounted to around 10 %. Under such dry conditions, SVA makes out a relevant part of diel evaporation suggesting its relevance to improve the quantification of land-atmosphere exchange at a sub-daily scale. Our results open the opportunity to get a conservative estimate of SVA at larger timescales. More comparisons with long-term measurements but also short-term sampling campaigns
near the EC footprint can provide valuable insights that are necessary to validate our findings. Lastly incorporating fully-coupled soil hydrological and land-surface modeling, considering the transport of water (in liquid and vapor form) and heat, similar to the approaches used by Sakai et al. (2009), Saaltink et al. (2020), and Garcia Gonzalez et al. (2012) will help in understanding the uncertainties related to lysimeter SVA measurements. By pursuing these avenues, we can significantly enhance our understanding of the field and pave the way for further discoveries.

| Symbol | Full form | Unit |
|---|---|---|
| $M_w$ | Molecular weight of water = 0.018 | $\mathrm{kg\,mol^{-1}}$ |
| $R$ | Universal gas constant = 8.314 | $\mathrm{J\,mol^{-1}\,K^{-1}}$ |
| $SWC$ | Volumetric soil water content | $\mathrm{m^3\,m^{-3}}$ |
| $T_a$ | Air temperature | °C |
| $T_s$ | Surface temperature | °C |
| $T_{dew}$ | Atmospheric dewpoint temperature | °C |
| $T_{soil}$ | Soil temperature | °C |
| $\rho_w$ | Density of water | $\mathrm{kg\,m^{-3}}$ |
| $F_{EC}$ | $H_2O$ flux measured with the EC method | mm per unit of time |
| $F_{IN,EC}$ | downwards directed $H_2O$ flux measured with Eddy Covariance technique | mm per unit of time |
| $F_{IN,LYS}$ | incoming/condensing $H_2O$ flux measured with lysimeter technique | mm per unit of time |
| $F_{IN}$ | downwards directed $H_2O$ flux (for EC) and incoming/condensing $H_2O$ flux (for lysimeter), respectively | mm per unit of time |
| $F_{LYS}$ | $H_2O$ flux measured with the lysimeter method | mm per unit of time |
| $F_{OUT,EC}$ | upwards directed $H_2O$ flux (for EC) and outgoing/evaporating $H_2O$ flux (for lysimeter), respectively | mm per unit of time |
| $F_{OUT,LYS}$ | upwards directed $H_2O$ flux (for EC) and outgoing/evaporating $H_2O$ flux (for lysimeter), respectively | mm per unit of time |
| $F_{OUT}$ | upwards directed $H_2O$ flux (for EC) and outgoing/evaporating $H_2O$ flux (for lysimeter), respectively | mm per unit of time |
| $F$ | $H_2O$ flux | mm per unit of time |
| $LW$ | Long wave radiation | $\mathrm{W\,m^2}$ |
| $SW$ | Short wave radiation | $\mathrm{W\,m^2}$ |
| $\Psi_m$ | Soil matric potential | hPa |
| $\Psi_w$ | Total soil water potential, constituted of matric, chemical, and pressure potential. | hPa |

| Symbol | Full form | Unit |
|---|---|---|
| $\lambda E$ | Latent heat flux | $\mathrm{W\,m^{-2}}$ |
| $\rho$ | Mole fraction of water vapor in dry air | $\mathrm{mol\,mol^{-1}}$ |
| $\sigma$ | Boltzmann's constant = $5.67 \times 10^{-8}$ | $\mathrm{W\,K^{-4}\,m^{-2}}$ |
| $\varepsilon$ | Emissivity of grass cover = 0.99 | NA |
| $e_a$ | Actual vapor pressure of the atmosphere | hPa |
| $e_{soil}$ | vapor pressure of soil air (dermined with the Kelvin equation) | kPa |
| $pF$ | **P**ower of ten of the **F**ree energy of soil water, log10 of Soil water potential | hPa |
| $rH$ | Relative humidity | % |
| $u^*$ | Friction velocity | $\mathrm{m\,s^{-1}}$ |
| $u$ | Wind speed | $\mathrm{m\,s^{-1}}$ |
| $u^*_{thres}$ | Threshold estimate of the friction velocity above which turbulent mixing is assumed. Minimum $u^*$ above which respiration measurements reaches a plateau. If not specified, $u*_{thres}$ refers to the 50th percentile of the threshold distribution. | $\mathrm{m\,s^{-1}}$ |

## Appendix B: Equations

**Relative humidity of the air in the soil pore space** ($rH$, %) was calculated based on $\Psi_m$ measurements of the heat dissipation sensor and $T_{soil}$ at the depth of -0.1 m for each lysimeter column in ES-LMa* based on the Kelvin equation (Edlefsen et al., 1943):

$$rH = exp(\frac{0.01 \cdot \Psi_m \cdot M_w}{R \cdot (T_{soil} + 273.15) \cdot \rho_w})$$
(B1)

with $\Psi_m$ in hPa, as negative soil water potential, $M_w$ is the molecular weight of water (0.018 kg mol$^{-1}$), $R$ is the universal gas constant (8.314 J mol$^{-1}$ K$^{-1}$), and $\rho_w$ is the density of water (1000 kg m$^{-3}$).

**Surface temperature** ($T_s$, °C) was calculated from measurements of the radiometric tower

$$T_s = \sqrt[4]{\frac{1}{\sigma \cdot \varepsilon} \cdot [LW_{OUT} - (1 - \varepsilon)LW_{IN}]} - 273.15$$
(B2)

where $LW$ is downwelling (LW$_{IN}$) and upwelling (LW$_{OUT}$) long wave radiation (W m$^{-2}$), $\sigma$ is Boltzmann's constant (W K$^{-4}$ m$^{-2}$) and $\varepsilon$ is emissivity of grass (-) and estimated to be 0.99. Note that this equation is less sensitive to $\varepsilon$ compared to the equation form that doesn't include LW$_{IN}$ (Thakur et al., 2022).

**Dewpoint temperature** ($T_{dew}$, °C) was calculated from $rH$ and $T_a$ based on the Magnus equation ($\lambda$ = 17.62, $\beta$ = 243.12) (Sonntag, 1990):

$$T_{dew} = \frac{\lambda \cdot \left(ln\left(\frac{rH}{100}\right) + \frac{\beta \cdot T_a}{\lambda + T_a}\right)}{\beta - \left(ln\left(\frac{rH}{100}\right) + \frac{\beta \cdot T_a}{\lambda + T_a}\right)}$$
(B3)

where $rH$ is relative humidity (%) and $T_a$ is air temperature (°C).

**Precision** and **recall** were calculated to compare the temporal consistency of the flux direction:

$$precision = \frac{tp}{tp + fp}$$
(B4)

$$recall = \frac{tp}{tp + fn}$$
(B5)

where *tp - true positives* are in the case of this study the number of observations where the EC method detects a $F_{IN,EC}$ simultaneously with i) at least one and ii) more than 50% of the lysimeters detecting $F_{IN,LYS}$. *fp - false positives* are observations of $F_{IN,EC}$ where lysimeters detect $F_{OUT,LYS}$, and *fn - false negatives* are observations of $F_{OUT,EC}$ while lysimeters detect $F_{IN,LYS}$.

**Appendix C: Predictor variable list**

Table C1: List of predictor variables used to model the difference between lysimeter and EC observations of $F_{IN}$; some variables were given in addition to the halfhourly measurement interval in the form of a rolling average over 24h (24h) or normalized by the range of observations of each sensor (norm)

| Category | Variable | Full form | variation |
|---|---|---|---|
| atmosphere | $e_a$ | Actual vapor pressure of the atmosphere | |
| | Wind direction | - | |
| | $u$ | Wind speed | |
| | $T_a$ | Air temperature | |
| | $rH$ | Relative humidity of the atmosphere | |
| | $\Delta T_s T_a$ | Difference between surface and air temperature | |
| | $\Delta T_s T_a$ | difference between the surface temperature and the air temperature | |
| eddy covariance | $u^*$ | Friction velocity | |
| | $x_{peak}$ | Along-wind distance providing the highest (peak) contribution to turbulent fluxes | |
| | $x_{offset}$ | Along-wind distance providing $\leq 1\%$ contribution to turbulent fluxes | |
| | EBC diff HH in MJ | diel difference of Energy Balance Closure in Megajoules | |
| | EBC diff HH | halfhourly difference of Energy Balance Closure | |
| | $LE_{scf}$ | Spectral correction factor for latent heat flux | |
| lysimeter | LYS $SWC$ | soil moisture at 0.1 m depth | norm, 24h |
| | LYS $pF$ | soil $\Psi_m$ at 0.1 m depth | norm, 24h |
| | LYS $rH_{SOIL}$ | relative humidity of the soil air (determined with the Kelvin equation) | norm, 24h |

| | LYS $e_{soil}$ | vapor pressure of soil air (determined with the Kelvin equation) | norm, 24h |
|---|---|---|---|
| | LYS $T_{soil}$ | soil temperature | |
| | $\Delta$ LYS $e_{soil}$ $e_a$ | difference between the vapor pressure of soil air and the atmosphere | norm, 24h |
| | $\Delta$ LYS $T_{soil}$ $T_{soil}$ | difference between the soil temperature within and outside the lysimeters | |
| | L[1:6] | Lysimeter ID (L1, L2, L3, L5, L6) as a categorial variable to account for a potential lumped effect of all static variables within each individual lysimeter, such as clay or soil organic carbon content. Only provided in *model.v2*. | |
| soil | $T_{soil}$ | soil temperature | |
| | $SWC$ | soil water content | norm, 24h |
| | $\Delta$ $T_{soil}$ $T_{soil}$ | difference between soil temperature at different depths | |

 **Appendix D: Diurnal EC and lysimeter measurements during SVA**

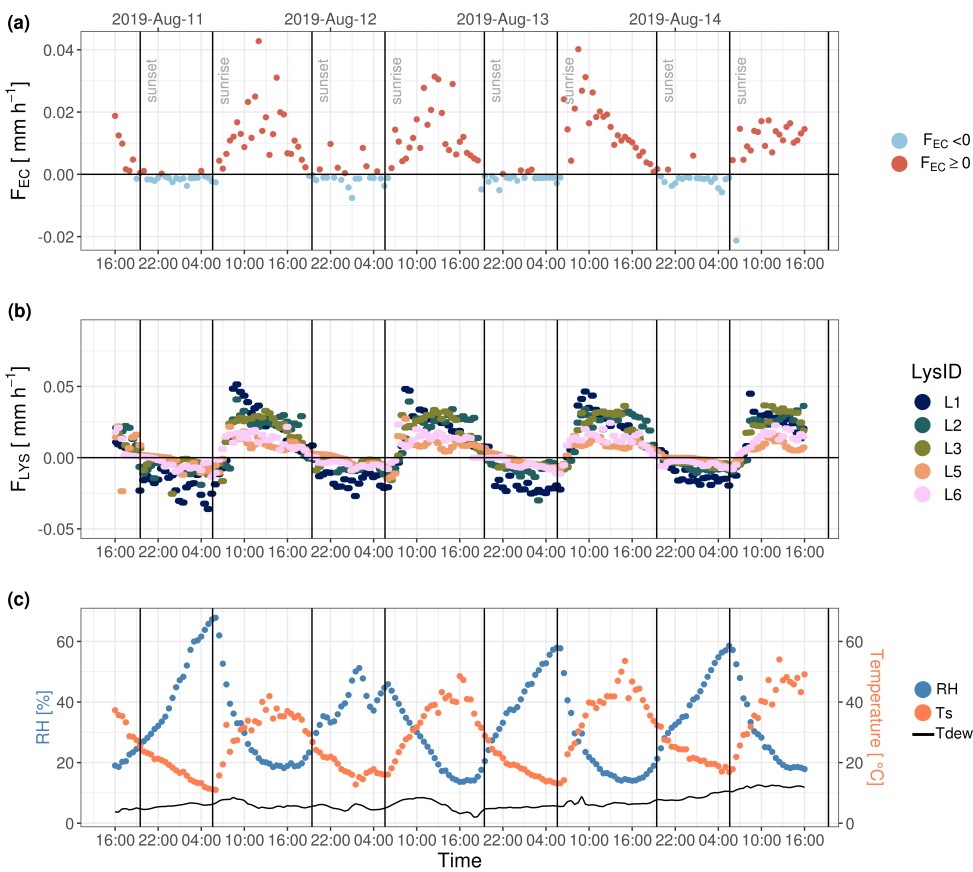

**Figure D1.** Diurnal measurements of water fluxes from (a) eddy-covariance ($F_{EC}$) and (b) the five lysimeters (L1, L2, L3, L5, and L6) from 11.08.2019 18:00 h until 15.08.2019 18:00 at ES-LMa*. Panel (c) illustrates the course of relative humidity (RH) at 2 m height above the soil surface together with surface ($T_s$) and dewpoint temperature ($T_{dew}$). Black vertical lines illustrate sunset and sunrise (determined by the geographic coordinates of the field site).

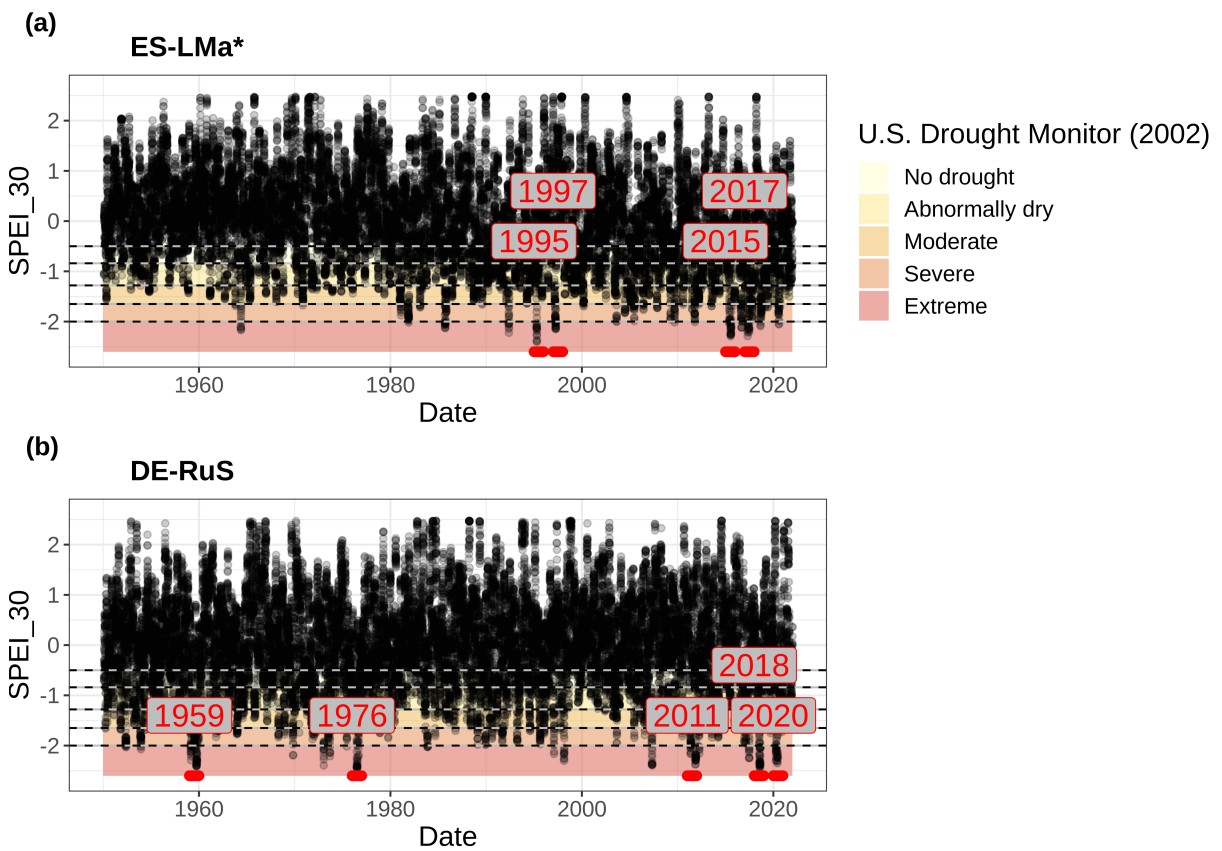

**Figure E1.** Standardized Precipitation Evaporation Index aggregated over 30 days (SPEI_30) from 1950 until 2022 for (a) Majadas de Tiétar (ES-LMa\*) and (b) Selhausen (DE-RuS) field site. The years with more than 2 weeks of extreme drought, as classified by the U.S. drought monitor (Svoboda et al., 2002), are highlighted by the red points and labels for each site, respectively.

## Appendix F: Influence of u* on flux direction

**Table F1.** *u\* threshold (m s$^{-1}$) estimates in ES-LMa\* and DE-RuS per year and season. At each site, periods of different surface roughness are considered with a and b representing dry and wet season in Majadas, and unplanted and planted conditions in Selhausen, respectively.*

| Site | Percentile | 2018 a | 2018 b | 2019 a | 2019 b | 2020 a | 2020 b | 2021 a | 2021 b |
|---|---|---|---|---|---|---|---|---|---|
| | | | | | [m s$^{-1}$] | | | | |
| ES-LMa* | u*$_{thres,05}$ | 0.052 | 0.055 | 0.055 | 0.055 | 0.055 | 0.050 | 0.055 | 0.050 |
| | u*$_{thres,50}$ | 0.076 | 0.073 | 0.073 | 0.073 | 0.073 | 0.069 | 0.067 | 0.069 |
| | u*$_{thres,95}$ | 0.098 | 0.098 | 0.098 | 0.103 | 0.103 | 0.090 | 0.090 | 0.083 |
| DE-RuS | u*$_{thres,05}$ | 0.054 | 0.056 | 0.056 | 0.055 | 0.055 | 0.055 | n.a. | n.a. |
| | u*$_{thres,50}$ | 0.069 | 0.068 | 0.068 | 0.093 | 0.093 | 0.093 | n.a. | n.a. |
| | u*$_{thres,95}$ | 0.106 | 0.140 | 0.140 | 0.177 | 0.177 | 0.177 | n.a. | n.a. |

**Table F2.** *Comparison of the number of simultaneous observations of flux direction towards/into the soil between EC and lysimeters for different u\* thresholds for ES-LMa\* and DE-RuS.*

| Site | Filter | | LE $< 0$ + meteo + | | |
|---|---|---|---|---|---|
| | | | u* $\geq 0.01$ | u* $\geq$ u*$_{thres,05}$ | u* $\geq$ u*$_{thres,95}$ |
| ES-LMa* | n night | | 445 | 425 | 304 |
| | n halfhours | | 3085 | 2278 | 1184 |
| | | 0 | 422 (13.7 %) | 303 (13.3%) | 147 (12.42%) |
| | n SVA halfhours | 3 | 2041 (66.2 %) | 1547 (67.6%) | 789 (66.6 %) |
| | | 5 | 829 (26.9 %) | 638 (27.9 %) | 307 (25.9%) |
| DE-RuS | n night | | 58 | 40 | 17 |
| | n halfhours | | 175 | 126 | 50 |
| | | 0 | 107 (61.1%) | 75 (59.5%) | 31 (62.0 %) |
| | n SVA halfhours | 3 | 26 (14.9%) | 24 (19.0%) | 10 (20.0 %) |

**Table F3.** Statistics for the comparison of $F_{IN,EC}$ and $F_{IN,LYS}$ as nighttime sums in ES-LMa* with different u* filtering thresholds.

| Flux direction | Filter | n | R | RMSE | MAE | intercept | slope | $r^2$ |
|---|---|---|---|---|---|---|---|---|
| | | | | | | [mm/ night] | | |
| night | $u^* \geq 0.01$ | 535 | 0.632 | 0.149 | 0.091 | 0.042 *** | 0.403 *** | 0.399 |
| | $u^*_{thres,05}$ | 530 | 0.651 | 0.133 | 0.077 | 0.037 *** | 0.446 *** | 0.423 |
| | $u^*_{thres,95}$ | 467 | 0.683 | 0.114 | 0.061 | 0.031 *** | 0.522 *** | 0.467 |
| night + $F_{IN,EC}$ | $u^* \geq 0.01$ | 445 | 0.266 | 0.081 | 0.050 | -0.031 *** | 0.150 *** | 0.071 |
| | $u^*_{thres,05}$ | 362 | 0.406 | 0.056 | 0.031 | - 0.015 *** | 0.189 *** | 0.165 |
| | $u^*_{thres,95}$ | 238 | 0.457 | 0.046 | 0.026 | - 0.011 *** | 0.284 *** | 0.209 |
| night + $F_{IN}$ | $u^* \geq 0.01$ | 130 | 0.663 | 0.033 | 0.024 | -0.002 *** | 0.492 *** | 0.440 |
| | $u^*_{thres,05}$ | 120 | 0.660 | 0.030 | 0.022 | - 0.002 *** | 0.515 *** | 0.435 |
| | $u^*_{thres,95}$ | 82 | 0.737 | 0.025 | 0.016 | - 0.002 *** | 0.579 *** | 0.543 |

**Appendix G: Timing of adsorption**

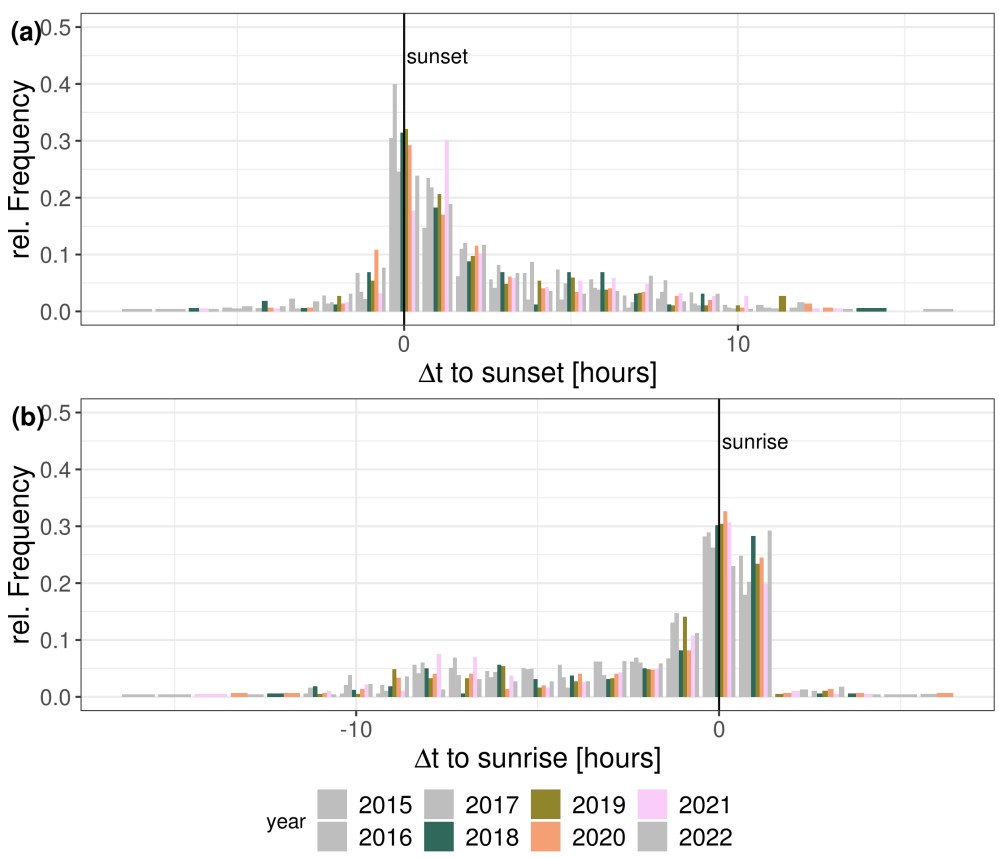

**Figure G1.** Relative frequency of (a) the first and (b) the last negative latent heat flux relative to sunrise and sunset, respectively, for the dry periods within 2015 to 2022 at the Majadas de Tiétar experimental field site. Note that since the dry periods deviate annually, the frequency of the timing is shown relative to the total number of dry days per year.

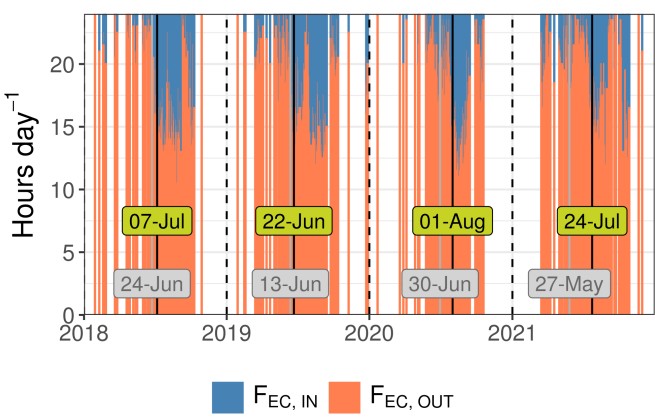

**Figure G2.** Illustration of diel fraction of positive (red) and negative (blue) $\lambda E$ fluxes measured with the EC method in ES-LMa*. The solid vertical lines mark the onset of adsorption dominated nights, defined as the first period each year, where five consecutive days with more than four hours of negative latent heat fluxes were observed. Black lines and green labels are based on EC method and grey lines with grey labels are based on lysimeter observations, respectively.

## Appendix H: Scatterplot and statistics DE-RuS

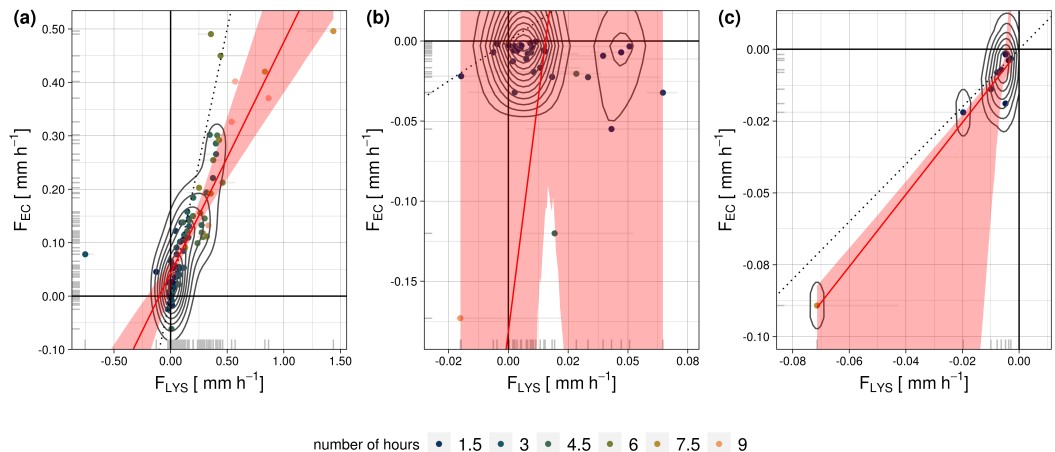

**Figure H1.** Comparison between night-time sums of lysimeter measured water fluxes ($F_{LYS}$) against eddy-covariance measured fluxes ($F_{EC}$) in Selhausen (DE-RuS) for different subsets of the data: (a) all good quality nighttime fluxes, (b) negative EC nighttime fluxes, and (c) negative nighttime fluxes and all lysimeter fluxes classified as soil adsorption of atmospheric vapor. The red line illustrates a major axis regression model and the red shading the confidence interval of the model. The black dotted line illustrates identity. Horizontal grey lines illustrate the minimum and maximum sum observed from single lysimeter columns. The colorcode illustrates the number of hours over which this sum was formed. It depends on how many observations were measured for the respective conditions on each night.

**Table H1.** Statistics for the comparison of $F_{IN,EC}$ and $F_{IN,LYS}$ as nighttime sums in DE-RuS with different filtering periods. *See also Fig. H1*

| Site | Filter | n | R | RMSE | MAE | intercept | slope | $r^2$ |
|------|--------|---|---|------|-----|-----------|-------|-------|
| | | | | | | [mm/ night] | | |
| | night | 91 | 0.816 | 0.170 | 0.083 | 0.043 *** | 0.432 *** | 0.666 |
| DE-RuS | night + $F_{IN,EC}$ | 31 | 0.115 | 0.052 | 0.035 | -0.182 n.s. | 11.650 n.s. | 0.013 |
| | night + $F_{IN}$ | 4 | 0.964 | 0.002 | 0.002 | 0.000 ** | 1.254 ** | 0.968 |

 **Appendix I: Scatterplot individual lysimeters ES-LMa**

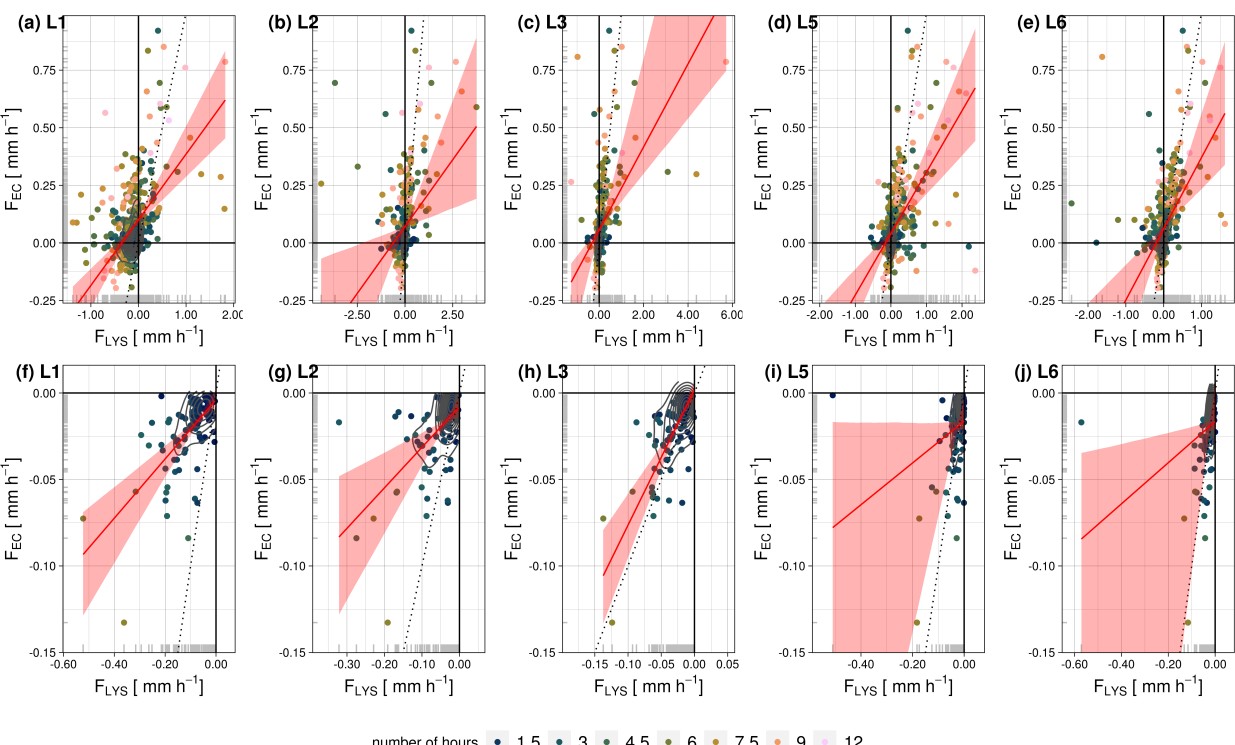

**Figure I1.** Comparison between night-time sums of lysimeter measured water fluxes ($F_{LYS}$) against eddy-covariance measured fluxes ($F_{EC}$) in Las Majadas de Tiétar (ES-LMa*) for the individual lysimeters (L1, L2, L3, L5, L6) and for different subsets of the data: *toprow:* all good quality $u^*$-filtered nighttime fluxes, *bottomrow:* good quality $u^*$-filtered negative nighttime fluxes and all lysimeter fluxes classified as soil adsorption of atmospheric vapor. The red line illustrates a major axis regression model and the red shading the confidence interval of the model. The black dotted line illustrates identity. The colorcode illustrates the number of hours over which this sum was formed. It depends on how many observations were measured for the respective conditions on each night.

**Table I1.** Statistics for the comparison of $F_{IN,EC}$ and $F_{IN,LYS}$ as nighttime sums for each individual lysimeter column in ES-LMa* for different subsets of the data: 1. all good quality $u^*$-filtered nighttime fluxes and 2. negative nighttime fluxes and all lysimeter fluxes classified as soil adsorption of atmospheric vapor. *See also Fig. I1*

| Site | LysId | Filter | n | R | RMSE | MAE | intercept | slope | $r^2$ |
|------|-------|--------|---|---|------|-----|-----------|-------|-------|
| | | | | | | | [mm/ night] | | |
| ES-LMa* | L1 | night | 548 | 0.468 | 0.319 | 0.223 | 0.101 ** | 0.285 ** | 0.219 |
| | | night + $F_{IN}$ | 108 | 0.659 | 0.103 | 0.075 | -0.004 ** | 0.171 ** | 0.435 |
| | L2 | night | 547 | 0.349 | 0.460 | 0.193 | 0.071 ** | 0.116 ** | 0.122 |
| | | night + $F_{IN}$ | 107 | 0.578 | 0.057 | 0.035 | -0.008 ** | 0.234 ** | 0.334 |
| | L3 | night | 349 | 0.496 | 0.451 | 0.158 | 0.057 ** | 0.180 ** | 0.246 |
| | | night + $F_{IN}$ | 108 | 0.727 | 0.019 | 0.013 | 0.002 ** | 0.784 ** | 0.528 |
| | L5 | night | 548 | 0.523 | 0.309 | 0.147 | 0.041 ** | 0.267 ** | 0.273 |
| | | night + $F_{IN}$ | 108 | 0.283 | 0.055 | 0.021 | -0.016 . | 0.121 . | 0.080 |
| | L6 | night | 550 | 0.474 | 0.262 | 0.113 | 0.061 ** | 0.309 ** | 0.225 |
| | | night + $F_{IN}$ | 108 | 0.289 | 0.056 | 0.017 | 0.017 . | 0.119 . | 0.084 |

**Appendix J: Proportion of random error on $F_{IN,EC}$**

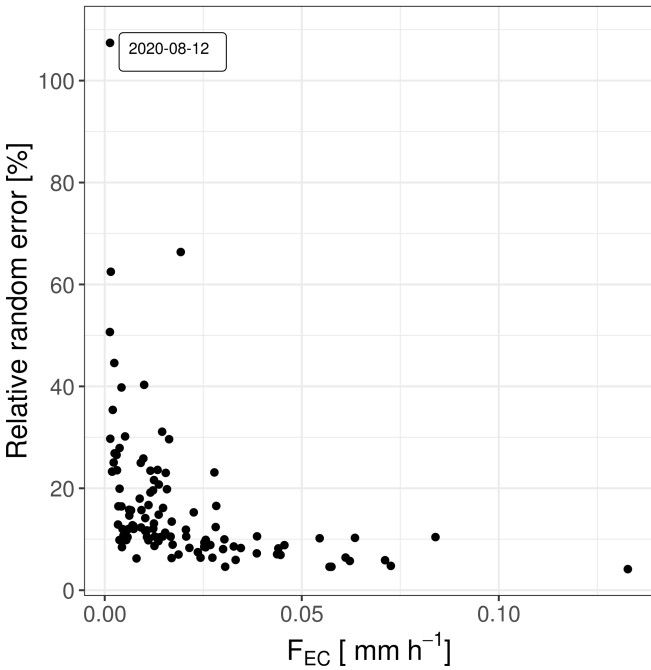

**Figure J1.** The relative random error shown on the y-axis is the proportion of the random error from the total inward flux measurements $F_{IN,EC}$ of the EC for each nights. The half hourly $F_{IN,EC}$ measurements per night were summed. The random error per night was determined by propagating the random error of the half houly measurements using standard deviations.

## Appendix K: Modeling results with given Lysimeter ID

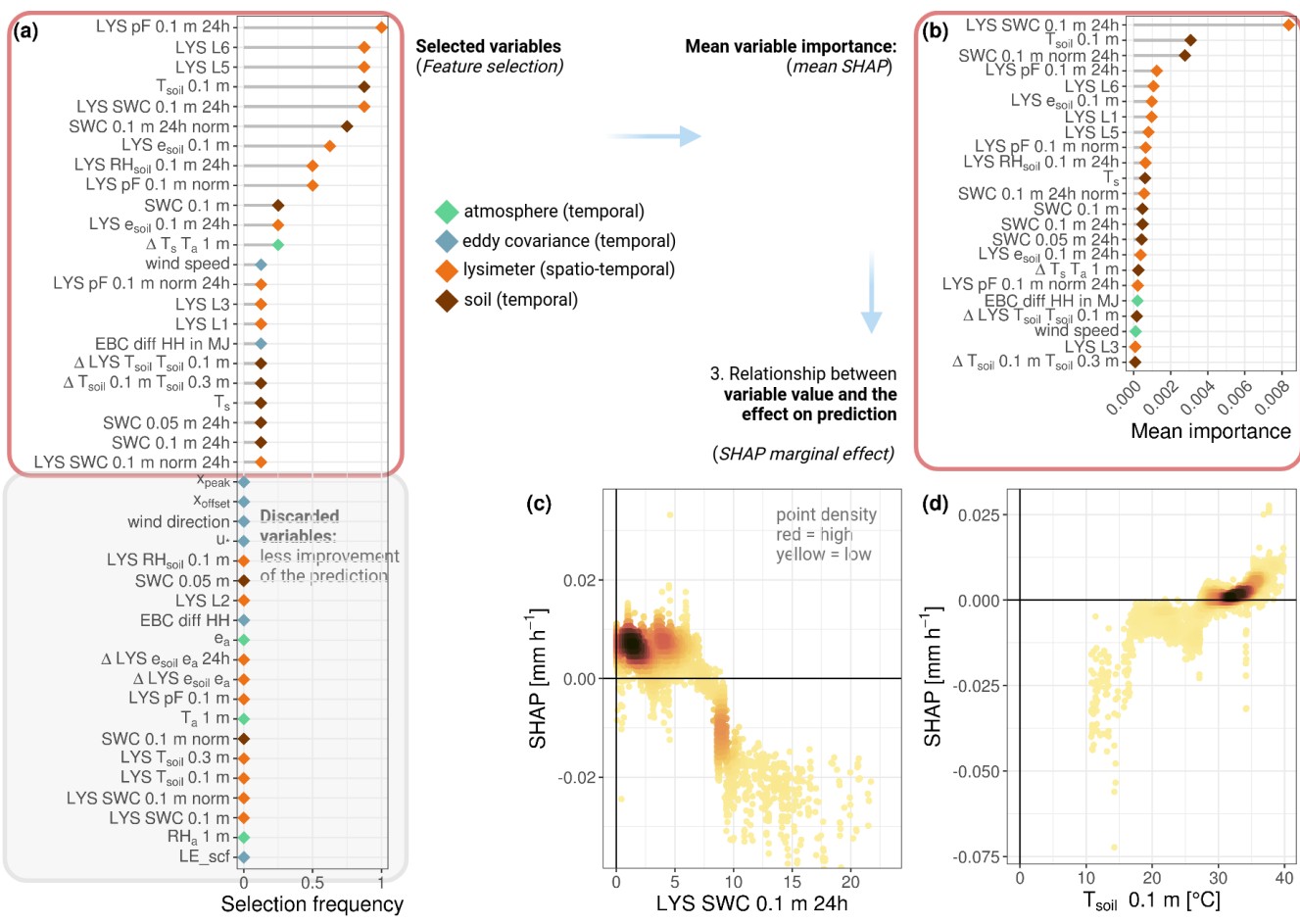

**Figure K1.** Feature selection and variable importance with predictor variable set including lysimeter ID as additional information: (a) Selection frequency of predictor variables of the best models, (b) summary graph for variable importance from high to low, based on the ensemble mean SHAP value of each predictor variable, and half-hourly SHAP influence of single observations of the two most important predictor variables: (c) 24h-smoothed $SWC$ within lysimeters at 10 cm depth, and (d) soil temperature within lysimeters at 10 cm depth. A description of all predictor variables is given in Appendix C.

**Appendix L: Influence of u* on the mismatch between lysimeter and EC**

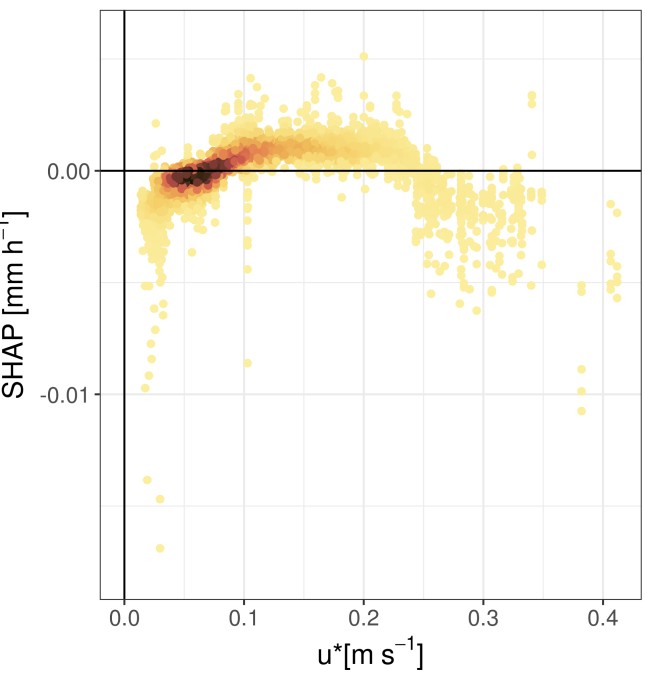

**Figure L1.** Impact of $u^*$ on the prediction of the half-hourly differences between lysimeters and EC observations, quantified with SHAP values across the range of observed $u^*$.

**Appendix M:  Diel ratio of incoming and outgoing water fluxes at ES-LMa\***

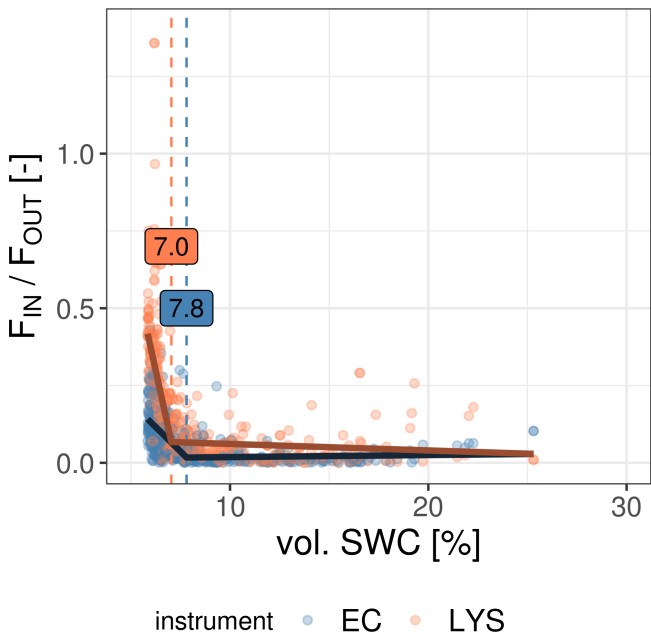

**Figure M1.** Daily ratio of $F_{IN}$ over $F_{OUT}$ across *in situ* soil water content ($SWC$) in Majadas de Tiétar measured with lysimeters (red) and the Eddy Covariance (EC) method (blue). The vertical dashed lines illustrate the *breakpoint* identified with a segmented linear regression independently for each measurement method.

*Code and data availability.* Data and the R code to reproduce the results of this analysis can be shared on request.

*Author contributions.* MM, RO, MR, and SP designed the setup and planned the study. GM, AC, TEM, and MM maintained the site ES-LMa* and the instrumentation and conducted the measurements. SP, and TEM processed the data from ES-LMa*. SP analyzed the data and prepared the original draft, both under the supervision of MM, RO, AH, and SCL. JG and AG, together with the acknowledged staff, performed the site set-up and operation, data processing, and initial quality control for DE-RuS. All authors discussed, reviewed, and edited
the paper. Sinikka J. Paulus wishes to thank Emmanuel Arthur for sharing his knowledge on estimating soil mineralogy using soil sorption isotherms.

*Competing interests.* None of the authors has any competing interests.

*Acknowledgements.* We thank the city council of Majadas de Tiétar, for support. Our thanks go to Martin Strube, Ramón López-Jimenez, Martin Hertel, and the Freiland group at the MPI-BGC, for great technical and scientific assistance during fieldwork. AG and JG thank Marius
Schmidt, Daniel Dolfus, Werner Küpper, and Philipp Meulendick for instrument operation and data processing at the site DE-RuS. Sinikka Jasmin Paulus has been supported by funding through the International Max Planck Research School for Global Biogeochemical Cycles (IMPRS-gBGC) at the University of Jena. René Orth has been supported by funding from the German Research Foundation (Emmy Noether Grant; grant no. 391059971). The Alexander von Humboldt Foundation has supported this research through the Max Planck Research Prize 2013 to Markus Reichstein. Jannis Groh is funded by the Deutsche Forschungsgemeinschaft (DFG, German Research Foundation) - project
no. 460817082.

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
