# Peer review of "Interpretability of negative latent heat fluxes from Eddy Covariance measurements during dry conditions"

_EGUsphere, 2023_

## Referee Comment (RC1)

**Major comments**

This is a very interesting and well written paper that focusses on the capacity of EC systems to detect water vapour adsorption (using lysimeter data as verification data), for 2 contrasting sites/climates. The authors present solid data sets, further strengthened by careful filtering, and statistical analyses (including machine learning methods).

While I think it is a good thing that more than 1 site is presented, in my opinion the use of a temperate site (with very little adsorption) is a bit of a distraction to the main story line. However, note that I am not suggesting removing these data and related discussion, unless the other reviewers are of a similar opinion.

There are two shortcomings (although not 'deal-breakers' for me recommending this paper to be accepted pending minor revisions):

- there is little discussion of the (spatiotemporal variation in) EC footprint, which must have played an important role for the German site in particular

- the soil physical properties that affect the adsorption isotherm (clay type and clay content) are not mentioned (nor are data provided to describe the water retention curve). I think some of these properties (clay amount) could help explain the variation among the lysimeters.

**Minor comments**

- Line 1: In my opinion SVA also occurs during the daytime (mid to late afternoon), especially when there is a moist sea-breeze coming in.
- Line 24/25: You say "Although the two instruments substantially differ with regard to the evaporative fraction with 64 % and 25 % for the lysimeter and EC methods, they are in either case substantial". Is this for the full 24 hours? Or for nighttime only? And how is EF defined here?
- Line 28: "The adsorption of atmospheric water vapor by dry soils (SVA)". What does the S in SVA stand for? Soil? So Soil Vapor Adsorption. Make this clearer.
- Line 42: Replace 'soil parameters' by 'soil variables'
- Line 44-46: "The model-based numerical evaluation further confirmed the ability of this type of lysimeters to quantify SVA correctly (Saaltink et al., 2020). Based on the long time series, SVA has been observed to reach significant magnitudes". You are overusing the word 'the'. Say instead: " Model-based numerical evaluations further confirmed the ability of this type of lysimeters to quantify SVA correctly (Saaltink et al., 2020). Based on analysis of long time-series, SVA has been observed to reach considerable magnitudes".
- Line 55: Brackets missing around Kool et al. (2021)
- Line 58: What is meant by "SVA observations"? Do you mean independently verified 'by eye'? Surely EC measurements are also observations?
- Line 64: You mean "for all **vapour** fluxes at the land's surface"?

- Line 72: You could mention the word 'footprint' here and refer to the relevant EC literature?
- Line 92: say" "vertical distance between the EC sensors and the adsorbing soil surface"
- Line 99-100: I would remove "However, readers who are already well-versed in this subject matter may choose to skip these sections if deemed unnecessary". Most people already read scientific papers in a selective way.
- Line 101: I would say: "is expressed as the deviation of the total potential energy of the soil water relative to the reference state"
- Line 103: say: " .. is generally expressed in units of pressure...". I say this because of course it can be expressed in different units. On a mass basis we work with energy (Joules) per kg. Often J m$^{-3}$ are used (potential on volume basis). This has the advantage that J m$^{-3}$ = N m$^{-2}$ = Pa.
- Line 103-104 are somewhat clumsy. Why are you comparing and contrasting SWC and psi_w in this way?
- Line 106: if you talk about the dominant force being the matric potential, then you may as well introduce the gravitational potential higher up, for completion and to avoid confusing the not so well versed readers..
- Line 112-113: you say "For better understanding, we added the conversion of Ψw into the SWC of a loamy sand (van Genuchten, 1980)". This is an akward way of saying that you used the water retention curve of a typical loamy sand to derive the SWC from Ψw values. Please keep the language as soil physical as possible.
- Line 118: say: " are adsorbed onto the soil particles".
- Line 129-140: It would be interesting to know what clay-type is found at this site (if will affect the shape of the adsorption isotherm) and how close to the sea this site is, e.g. are there sea-breeze effects that bring in moist air from the sea that gets adsorpted by the dry soil. I looked on the map and it is far away from the sea. Unlike the paper by Verhoef et al. that studied a site near Seville where moist air coming in in the late afternoon played a role.
- 142-152: again, do you now the clay type here?
- Section 3.2. Nowhere in this section do you mention the dominant wind direction and typical footprint size. I think this is an important omission. From googlemaps I can see that the area around the mediterreanean site is pretty homogeneous, but this is not the case for Selhausen? I am therefore not sure how comparable the lysimeter (bare soil) and EC data (vegetated field, representative of cropped fields overall which would decrease adsorption compare to open bare soil??) are for Selhausen? You say yourself later on "SVA was reported to be reduced below or in the vicinity of tall, active vegetation by 76 % (Kosmas).
- Also, in Section 3.2 and 3.3. Can you please give the resolution of the temperature and SWC sensors?
- Line 245: Put Sonntag in brackets
- Line 248: "downwelling (↓) and upwelling (↑)". Are the symbols SW and LW missing here? This looks strange.
- Starting at line 284: Would it not be better to use upward and downward arrows instead of IN and OUT?

- Line 305: This is the first time that you mention the words 'soil hydraulic conditions'. Should this not be introduced much higher up?
- Line 306: This predictor list is very comprehensive, but what is missing are proper representatives of soil hydraulic properties (e.g. the slope of the water retention curve or air-entry value, or even better, properties that denote the specific surface area of the adsorbing soil particles, the clay in particular. This seems an omission to me.
- Line 327: why would adsorption be highest when "soil water supply is high"? Surely the soil needs to be fairly dry for adsorption to occur? Did you mean $F_{OUT,EC}$?
- In Figures 3 and 4, make it clear that "solid vertical lines that mark the end of the night, sunrise, sunset, and beginning of the night, respectively" are in fact curved, not straight. Also, what is the straight vertical grey bar in Figure 4a??
- Line 360-361: Sure it depends on "soil-intrinsic physical properties", but you never actually say what these are. Also, it also depends on the soil variables (soil temperature and soil moisture content)s, so please add this?
- Line 369: replace "more dry" with "drier"
- In Figure 6: Has the number of hours colouring code been explained properly and is it actually used much in the description around this graph? I found it a bit confusing. Around line 445 you discuss it a bit but it is not clear to me how these number of hours were counted or selected?
- Line 435-436: I don't know what you mean by "This is not surprising given that under stable nighttime conditions F is suspected to leave the control volume other than in the vertical direction (advection, drainage flows) and thus undetected by the EC sensor". Is there a word missing ("not" in front of "suspected")?
- Following on from this: You have not defined control volume, footprint, fetch etc. higher up. In fact a footprint analysis for the data would be very welcome.
- Around Figure 8. The discussion here focusses on the temporally variable soil variables Ts and SWC. But it is not more likely that modest variations in clay content (and SOM?) among these lysimeters affect the adsorptive capacity of the soil in these lysimeters? Verhoef et al. (and Kosmas?) showed that adsorption is quite sensitive to clay content?
- Around Figure 8. There is no real explanation/discussion of the SHAP plots c and d. I am not sure how to interpret these and what it all means.
- Lines 576 and 577: incorporating vapour fluxes is not just important for hydrological models but also for land surface models. It would be good to make that point here and add the following reference? Garcia Gonzalez, R. , Verhoef, A. , Vidale, P. , Braud, I. (2012) Incorporation of water vapour transfer in the JULES Land Surface Model: implications for key soil variables and land surface fluxes. Water Resources Research , 48 (5). ISSN: 0043-1397 | doi: https://dx.doi.org/10.1029/2011WR011811

---

## Author Comment (AC1)

| DOI | https://doi.org/10.5194/egusphere-2023-2556 |
|---|---|
| Interpretability of negative latent heat fluxes from Eddy Covariance measurements during dry conditions | |
| Author(s) | Sinikka J. Paulus et al. |
| Handling Editor | Andrew Feldman |
| Manuscript type | Research article |
| Status | Final response (author comments only) |

**Response to Report from Reviewer #1**

Reviewer comments are printed in black.
Answers are printed in blue below the respective comment.
* * *
**Major comments**

This is a very interesting and well written paper that focusses on the capacity of EC systems to detect water vapour adsorption (using lysimeter data as verification data), for 2 contrasting sites/climates. The authors present solid data sets, further strengthened by careful filtering, and statistical analyses (including machine learning methods).

Thank you very much for this assessment and for the very helpful comments. We considered all of them in detail and they have helped us substantially in improving the quality of this manuscript from various perspectives. Please find below a point-by-point reply to your comments and suggested changes in the revised manuscript

While I think it is a good thing that more than 1 site is presented, in my opinion the use of a temperate site (with very little adsorption) is a bit of a distraction to the main story line. However, note that I am not suggesting removing these data and related discussion, unless the other reviewers are of a similar opinion.

We thank the reviewer for this suggestion. Although we follow the reasoning of the reviewer's suggestion, we decided to keep the Selhausen results in the main manuscript: We think that results from a temperate site where the process of soil water vapor adsorption is unlikely to happen are informative to have in the manuscript. This shows that our approach does not necessarily indicate adsorption and agreement between EC and lysimeters in any case (falsification test), and, more importantly, illustrates that adsorption can become relevant in ecosystems in the context of climate change inducing more intense and prolonged droughts. Moreover, we note that the majority of the main figures focus on the results from the Mediterranean site, in line with the suggestion of the reviewer to highlight and exploit these data more.  Finally, the other reviewer has not expressed similar thoughts.

There are two shortcomings (although not 'deal-breakers' for me recommending this paper to be accepted pending minor revisions):

- there is little discussion of the (spatiotemporal variation in) EC footprint, which must have played an important role for the German site in particular

*The reviewer raises a relevant point. Footprint analysis aiming to evaluate whether half-hourly flux values are sufficiently representative of the target area has already been performed for both sites based on the footprint model by Kluijun et al. 2015. The analysis confirmed for both sites a cumulative contribution of > 80 % being from the target area. We added the visualization of the 2D-Footprint models for both sites into Figure 2, where the setup is presented.*

*The analysis confirmed that all of the measured flux stemmed from within the target area, which is in Selhausen (Fig. 2 b) the agricultural field in which center the tower is located.*

[Figure]

*Figure 2: Aerial image of (a) the Majadas de Tiétar (ES-LMa\*) and (b) the Selhausen agricultural field (DE-RuS) sites. The squares show the location of Eddy Covariance (EC) instruments (light blue) and the lysimeters (red) at each site. Note that the spatial resolution differs. The EC footprint climatology isolines are overlaid in grey (for 50%, 70%, and 80% of the climatology, respectively; EC footprint of Selhausen is retrieved from ICOS (2021)). (Map data from Google Earth; (a) image from Instituto Geográfico Nacional, (b) image from GeoBasis-DE/BKG).*

*We added the information about the footprint model and relevant literature to all the sections and lines, where the reviewer pointed out this information was missing (line 72, Section 3.2, and lines 435-436).*

**Introduction**:

*"Also, EC measures the turbulent vertical transport of gases at some meters above the soil surface whereas lysimeters measure the phase change of water (vapor ⇄ liquid or solid) at the ground level. Another difference between lysimeters and EC is that the size, shape, and position of the surface area of influence and vary for EC depending on the wind direction and turbulence conditions (Amiro, 1998; Schmid 2001, Schmid 1994, Kljun et al. 2015), whereas lysimeters are spatially stationary and always measure the same volume of soil."*

**Section 3.2:**

*"2D-Footprint analysis aiming to evaluate whether half-hourly flux values are sufficiently representative of the target area were performed for both sites based on the model by Kluijun et al. 2015 (illustrated as footprint climatology isolines in Fig. 2; ES-LMa: from 2015-2017; in Selhausen: 2018-2019, more details given in ICOS (2021)). In ES-LMa\* the 80% footprint climatology is within a distance of 33 m from the tower in the two dominant wind directions. In Selhausen, 80% of footprint climatology is within the agricultural field in the dominantly prevailing west-south-west wind direction."*

**lines 435-436:**

*"One interpretation of this result could be that each lysimeter covers a smaller spatial scale ($1\ m^2$ each) compared to the EC (illustrated in Figure 2 as footprint climatology) but the average across lysimeters is a better representation of the spatial mean and is therefore more in line with the EC observations.*

*Nevertheless, a structural difference between the measuring instruments in the form of a bias remains. This is not surprising given that under stable nighttime conditions the ratio between the vertical and the non-vertical (drainage and advection) movement of F is suspected to be smaller. As a result, a larger proportion of the total F leaving the source area remains undetected by the EC sensor than in measurements during the day with good atmospheric mixing (Wohlfahrt et al., 2005). "*

Amiro, B. D. (1998). Footprint climatologies for evapotranspiration in a boreal catchment. Agricultural and Forest Meteorology, 90(3), 195-201.

ICOS (2021). DE-RuS ICOS Ecosystem Station Labelling Report. Carbon Portal. https://hdl.handle.net/11676/HE7Nj8_yV0xTr1zYLvOchSR1

Kljun, N., Calanca, P., Rotach, M. W., & Schmid, H. P. (2015). A simple two-dimensional parameterisation for Flux Footprint Prediction (FFP). *Geoscientific Model Development*, *8*(11), 3695-3713.

Schmid, H. P. (2002). Footprint modeling for vegetation atmosphere exchange studies: a review and perspective. Agricultural and Forest Meteorology, 113(1-4), 159-183.

- the soil physical properties that affect the adsorption isotherm (clay type and clay content) are not mentioned (nor are data provided to describe the water retention curve). I think some of these properties (clay amount) could help explain the variation among the lysimeters.

The reviewer addresses two main points concerning soil physical properties i) that the clay amount within the lysimeters is not measured, hence it should be mentioned that the clay

amount could differ among lysimeters and cause the observed variation between lysimeters, and ii) that data of the (dominant) clay type (clay mineralogy) would be desirable.

We agree that the relevance of clay content, soil organic carbon content, as well as clay mineralogy, should be strongly highlighted in the revised version of this manuscript. Thank you very much for pointing this out, we learned a lot during the process of answering your comment.

**Regarding i)** unfortunately, we don't have measured clay content (or soil organic carbon) within each lysimeter. It is also not possible to obtain this data since it would require destructive sampling from the lysimeter columns. Therefore, we only reported the clay amount for the whole site in Majadas (5 %, preprint line 139) and measured within the EC footprint and in the surroundings of the lysimeters. In a recent campaign, however, soil texture was analyzed for topsoil samples and revealed that although the clay content was usually very low (around 5 %) one sample had a clay content of 18 %. Such changes affect small-scale soil water vapor adsorption behavior at our site and the reviewer was very right to point this out.

However, the effect of potential variations in the clay amount between lysimeters is already indirectly integrated into our modeling approach in section 4.4 and Appendix Figure H1. To assess the lumped effect of all static variables of each individual lysimeter (which includes differences in clay content) on the difference between Lysimeter and EC observations we included "LysID" as one predictor variable in model.v2. If this (lumped sum of) static effects would dominantly add to explain the differences between the instruments, it should show up in the SHAP values. However, it seems like there is more information content in the temporally variable surface temperature and soil water content.

Nevertheless, we agree with the reviewer that the differences in the lysimeter SWCs could be caused by variations in clay content, and it is a very important perspective that should be emphasized more in our manuscript. Therefore, we make this more clear in the discussion of the modeling results:

*"This means that the lumped effect of static properties which might deviate between lysimeters such as clay or organic matter carries a lower information content for the prediction of the differences between instruments."*

*"It is highly possible that the spatio-temporal differences in soil hydraulic conditions of the lysimeters are caused by small differences in soil properties such as clay or soil organic carbon content. Both variables are known to substantially increase soil sorption capacity and to generally affect soil water retention characteristics (Arthur et al. 2015, 2016). At the Majadas de Tiétar field site, the topsoil clay content is relatively constant between 0 and 5 % but an individual topsoil sample from outside the lysimeters contained 18 % clay, which is more similar to the clay content in the subsoil. Such outliers in the spatial distribution of clay content have substantial non-linear effects on small-scale variations in the soil water retention characteristics at the dry end of the water retention curve and thereby could cause the observed variability of SWC between lysimeters."*

Further, we realized, that we did not report the soil texture values (only soil classification) in Selhausen in the Material and Methods section and added this information to the respective sentence:

line 142 -152: *"The soil at the site is classified as a Cutanic Luvisoll (Pütz et al., 2016) and the soil texture of the different soil horizons (Ap, Al-Bv, II-Btv) can be classified according to USDA 2017 as silt loam (U.S. Department of Agriculture, 2017; Groh et al., 2020) with 15% sand,  68 % silt, and 17 % clay content (Groh et al., 2020)."*

**Regarding ii)** again, we do not have measurements of the Majadas site's mineralogy. We did a literature search on the clay mineralogy in the area.

Studies from Casatejada, a town approximately 10 km away from our study site ES-LMa measured dominantly Smectite (60%), followed by Illit (25 %) and Kaolinit (15%) (Instituto Geológico y Minero de España, 1992). A mixture of clay minerals with a high-swelling capacity clay being present in the soil is in line with adsorption occurring already at low RH (40%) in Majadas. Also, it explains the measured adsorption at a site with predominantly low clay content in the topsoil (5%). Unfortunately, we are not able to resolve this comment in further detail. However, we agree on the importance of better integrating the relevance of clay mineralogy on adsorption behavior in general in the revised version of the manuscript. The mineralogy of the Selhausen site was measured in a former study and determined to be predominantly illite with the presence of chlorite and/or vermiculite and minor amounts of kaolinite (Jiang et al. 2014).

We added the information on the clay mineralogy and at all points in the manuscript where it was suggested by the reviewer: to the site description of ES-LMa* line 129-140; to the site description of DE-RuS 142 -152; and in the specification of the "soil intrinsic properties" line 360-361. Further, we added a sentence to discuss this additional information in the results section where reasons for the potential differences between sites are discussed:

line 129-140: *"The regional clay mineralogy was identified as a blend of smectite (45%), illite (35%) and chlorite and/or kaolinite (20%) (Instituto Geológico y Minero de España, 1992)."*

line 142 -152: *"The clay mineralogy of the site has been identified as a mixture of predominantly illite with the presence of chlorite and/or vermiculite and minor amounts of kaolinite (Jiang et al. 2014)."*

line 360-361: *"It underscores that while the probability of occurrence of SVA is influenced by climate (i.e. more common in semi-arid and arid regions), it can also occur in more humid regions. This is because it depends on soil-intrinsic physical properties, such as texture (clay content, clay mineralogy, and organic carbon content) (Orchiston, 1956; Arthur et al. 2019, Yukselen Akoy, 2010), soil structure that affects vapor transport characteristics (i.e. soil diffusion coefficient), and can happen anywhere if the dynamic requirements like temperature and moisture gradients are met."*

Results: *"At the same SWC under controlled conditions, the soil from DE-RuS should theoretically have a similar or higher capacity to adsorb water than the soil in ES-LMa* due to its high clay content (17% compared to 5%) which influences the water sorption behavior more strongly than the mineralogy for mixed soils with low kaolinite content (Arthur et al.*

*2015). Hence, these effects of the soil properties do not come into play when the overall climatic conditions are too wet."*

Arthur, E., Tuller, M., Moldrup, P., Jensen, D. K., & De Jonge, L. W. (2015). Prediction of clay content from water vapour sorption isotherms considering hysteresis and soil organic matter content. European Journal of Soil Science, 66(1), 206-217.

Instituto Geológico y Minero de España, 1992. MAPA GEOLÓGICO DE ESPAÑA - NAVALMORAL DE LA MATA.
https://info.igme.es/cartografiadigital/datos/magna50/memorias/MMagna0624.pdf

Jiang, C., Séquaris, J. M., Wacha, A., Bóta, A., Vereecken, H., & Klumpp, E. (2014). Effect of metal oxide on surface area and pore size of water-dispersible colloids from three German silt loam topsoils. *Geoderma*, *235*, 260-270.

Orchiston, H. D. (1954). Adsorption of water vapor: II. Clays at 25 C. Soil Science, 78(6), 463-480.

Yukselen-Aksoy, Y., & Kaya, A. (2010). Method dependency of relationships between specific surface area and soil physicochemical properties. *Applied Clay Science*, *50*(2), 182-190.

**Minor comments**
Line 1: In my opinion SVA also occurs during the daytime (mid to late afternoon), especially when there is a moist sea-breeze coming in.

We agree with the reviewer and deleted "during the night" in line 1.

Line 24/25: You say "Although the two instruments substantially differ with regard to the evaporative fraction with 64 % and 25 % for the lysimeter and EC methods, they are in either case substantial". Is this for the full 24 hours? Or for nighttime only?
And how is EF defined here?

We agree that the information in this sentence was not precise enough for the reader to understand. We adjusted the sentence according to the questions of the reviewer.

*"Although the two instruments substantially differ with regard to the measured ratio of adsorption over evaporation over 24 hours with 64% and 25% for the lysimeter and EC methods, they are in either case substantial."*

Line 28: "The adsorption of atmospheric water vapor by dry soils (SVA)". What does the S in SVA stand for? Soil? So Soil Vapor Adsorption. Make this clearer.

We added the information as requested.

*"The adsorption of atmospheric water vapor by dry soils (Soil Vapor Adsorption - SVA) has in recent years been identified to be underrepresented in ecosystem research (Saaltink et al., 2020)."*

Line 42: Replace 'soil parameters' by 'soil variables'

We replaced the word as suggested.

Line 44-46: "The model-based numerical evaluation further confirmed the ability of this type of lysimeters to quantify SVA correctly (Saaltink et al., 2020). Based on the long time series, SVA has been observed to reach significant magnitudes". You are overusing the word 'the'. Say instead: " Model-based numerical evaluations further confirmed the ability of this type of lysimeters to quantify SVA correctly (Saaltink et al., 2020). Based on analysis of long time-series, SVA has been observed to reach considerable magnitudes".

Thank you very much for this suggestion, we changed the lines accordingly.

Line 55: Brackets missing around Kool et al. (2021).

Thank you for pointing this out, we added the brackets.

Line 58: What is meant by "SVA observations"? Do you mean independently verified 'by eye'? Surely EC measurements are also observations?

Thank you for pointing out this language error. We specified the sentence by adding: "alongside independent SVA measurements".

Line 64: You mean "for all vapour fluxes at the land's surface"?

Yes. Thank you we added this specification.

Line 72: You could mention the word 'footprint' here and refer to the relevant EC literature?

Thank you for this suggestion, we included the word footprint with relevant literature to this sentence.

Line 92: say" "vertical distance between the EC sensors and the adsorbing soil surface"

Thank you for this suggestion for a more precise formulation of the hypothesis, we added it like suggested.

Line 99-100: I would remove "However, readers who are already well-versed in this subject matter may choose to skip these sections if deemed unnecessary". Most people already read scientific papers in a selective way.

We removed the sentence according to the reviewers suggestion.

Line 101: I would say: "is expressed as the deviation of the total potential energy of the soil water relative to the reference state"

Thank you for this suggestion, we adjusted the sentence accordingly.

Line 103: say: " .. is generally expressed in units of pressure...". I say this because of course it can be expressed in different units. On a mass basis we work with energy (Joules) per kg. Often J m-3 are used (potential on volume basis). This has the advantage that J m-3 = N m-2 = Pa.

Thank you for this suggestion, we adjusted the sentence accordingly.

Line 103-104 are somewhat clumsy. Why are you comparing and contrasting SWC and psi_w *in this way*?

Thank you for your suggestion. We changed the respective lines and hope that it is now less clumsy but still easy to understand for people without a soil-hydrological background:

*"Hence, Ψw describes the energy requirements to change the phase state of water or to induce water transport. It is non-linearly related to SWC. For soils with the same SWC, Ψw can differ by an order of magnitude due to variations in soil physical properties (Or, Tuller, and Wraith, 2022)."*

Line 106: if you talk about the dominant force being the matric potential, then you may as well introduce the gravitational potential higher up, for completion and to avoid confusing the not so well versed readers.

Thank you for your suggestion. Our feeling is, that this section can easily become very confusing. Our aim in this section is to provide the minimum information necessary to understand the main process that this study is investigating for people of the Eddy Covariance community so we didn't go further into detail but we understand that we should mention the different components and hope that we could sufficiently address your request by changing the sentences to the following:

*"The total Ψw is expressed as the sum of solute, pressure, gravitational, and matric potential. In many cases, the matric potential (Ψm, hPa) is the dominant component of Ψw."*

If not, we kindly ask for a specification about what would be missing in this section in the context of the study.

Line 112-113: you say "For better understanding, we added the conversion of Ψw into the SWC of a loamy sand (van Genuchten, 1980)". This is an akward way of saying that you used the water retention curve of a typical loamy sand to derive the SWC from Ψw values. Please keep the language as soil physical as possible.

Thank you for your suggestion, we changed the sentence accordingly:

*"We used the water retention curve of a typical loamy sand to derive SWC from Ψw (van*

*Genuchten, 1980)."*

Line 118: say: " are adsorbed onto the soil particles".

Thank you for your suggestion, we changed the sentence accordingly.

Line 129-140: It would be interesting to know what clay-type is found at this site (if will affect the shape of the adsorption isotherm) and how close to the sea this site is, e.g. are there sea-breeze effects that bring in moist air from the sea that gets adsorpted by the dry soil. I looked on the map and it is far away from the sea. Unlike the paper by Verhoef et al. that studied a site near Seville where moist air coming in in the late afternoon played a role.

Thank you for your suggestion. You are right, the sea is 270 km away from the site and we added this information to the site description.
We already addressed the clay type in the major comments section.

142-152: again, do you now the clay type here?

The clay type is addressed in the major comments section.

Section 3.2. Nowhere in this section do you mention the dominant wind direction and typical footprint size. I think this is an important omission.

This question was addressed under the answers to the major comments section.

From googlemaps I can see that the area around the mediterreanean site is pretty homogeneous, but this is not the case for Selhausen? I am therefore not sure how comparable the lysimeter (bare soil) and EC data (vegetated field, representative of cropped fields overall which would decrease adsorption compare to open bare soil??) are for Selhausen? You say yourself later on "SVA was reported to be reduced below or in the vicinity of tall, active vegetation by 76 % (Kosmas).

We hope that we have convinced the reviewer by adding the footprint area of the Eddy covariance tower to Figure 2 that the dominant part of the source area (in the dominant wind direction) is within one single agricultural field with homogeneous management, with the management practices recorded and visualized in Figure 4e.

The point about the comparability between the lysimeter and EC measurement at Selhausen due to the distance between instruments and different management practices is nevertheless a valid point that we recognized/pointed out already several times in our manuscript. (line 202, line 291, line 391 f.)

Also, for this reason, we make a comparison in section 4.2 for periods where at both instrument locations' the soil is bare.
We know that the setup at our temperate site is not ideal but nevertheless, we think that the conclusions we draw are valid (SVA occurs only seldom and then is only partially detected

by the EC, probably because soil moisture remains high even under relatively extreme dryness at the site).

Also, in Section 3.2 and 3.3. Can you please give the resolution of the temperature and SWC sensors?

The resolution of the SWC and temperature sensors in ES-LMa are 0.1% SWC and 0.02 °C and the resolution of the SWC sensor in Selhausen is 0.1 % SWC, both values according to the manufacturer. Further, we realized that there was a miscommunication and the sensor measuring soil temperature outside the lysimeter is a PT-100 temperature sensor from Jumo.

We added the information to the manuscript in the respective Sections:

**For ES-LMa***
*"Within each column, SWC and soil temperature (Tsoil, °C) (UMP-1, Umwelt-Geräte-Technik GmbH) are measured at 0.1 m soil depth at a resolution of 0.1% SWC and 0.02 °C, according to the manufacturer."*

*"Tsoil (PT-100, Jumo, Fulda, Germany) and SWC (Delta-ML3, Delta-T Devices Ltd, Burwell Cambridge, UK) were measured outside the lysimeters at 0.05 m, 0.10 m, and 0.2 m soil depth at a resolution of 0.02 °C Tsoil and 0.1 % SWC."*

**For DE-RuS:**
*"SWC is measured within each lysimeter at a depth of 0.1 m below the surface with time domain reflectometry probes (CS610, Campbell Scientific, North Logan, UT, USA) at a resolution of 0.1 % SWC, according to the manufacturer."*

*"SWC was measured at 0.025 m depth at a resolution of 0.1 % SWC, according to the manufacturer (CS616, Campbell Scientific, Logan (Utah), USA).*

Line 245: Put Sonntag in brackets.

Thank you.

Line 248: "downwelling ($\downarrow$) and upwelling ($\uparrow$)". Are the symbols SW and LW missing here? This looks strange.

Thank you for pointing this out, we have changed the arrows to IN and OUT to be consistent in the manuscript in the symbol/acronym for the direction of radiation and water flux (next comment).

*"Short- (SW , W m−2) and longwave (LW , W m-2 ) downwelling ($SW_{IN}$, $LW_{IN}\downarrow$) and upwelling ($SW_{OUT}$, $LW_{OUT}\uparrow$) radiation of the herbaceous layer"*

Starting at line 284: Would it not be better to use upward and downward arrows instead of IN and OUT?

We have already discussed this issue in great detail with our co-authors and have come up with this solution, and we would be very happy if we could keep it the way it is.

Line 305: This is the first time that you mention the words 'soil hydraulic conditions'. Should this not be introduced much higher up?

We agree with the reviewer also on this point, so when we revised the manuscript we focused on trying to include the word "soil hydraulic conditions" earlier and more often in the manuscript in the Sections Introduction and Theoretical Background. If there are other sections where we should replace our wording, we would be grateful for more specific suggestions.

Line 306: This predictor list is very comprehensive, but what is missing are proper representatives of soil hydraulic properties (e.g. the slope of the water retention curve or air-entry value, or even better, properties that denote the specific surface area of the adsorbing soil particles, the clay in particular. This seems an omission to me.

Thank you for raising this issue. We would like to confirm that the model was not trained using (static) soil hydraulic properties, such as the slope of the water retention curve or the air entry value since these parameters are not available at the sites.

However, the list of predictors does include (dynamic) in-situ measurements of $\Psi m$ based on heat dissipation sensors. We believe that these measurements contain sufficient information about the hydraulic state of the soil in the lysimeters. Our confidence in this approach is further supported by the modeling results, which indicate that the hydraulic state of the soil is one of the four most important predictor variables.

Further, as already mentioned in our answer to the majors comments that the lumped effect of all static variables (including soil hydraulic properties) is indirectly included in model.v2 as the predictor variable "LysID". If this (lumped sum of) static effect would add dominantly to explain the differences, it should show up in the SHAP values. However, it seems that there is more information content in the temporal variables Ts and SWC.

We believe that our response effectively addresses this point of criticism.

Line 327: why would adsorption be highest when "soil water supply is high"? Surely the soil needs to be fairly dry for adsorption to occur? Did you mean FOUT,EC?

Yes, thank you for pointing out this error.

In Figures 3 and 4, make it clear that "solid vertical lines that mark the end of the night, sunrise, sunset, and beginning of the night, respectively" are in fact curved, not straight. Also, what is the straight vertical grey bar in Figure 4a??

We adjusted the Figure caption as suggested and found the respective error in our code which produced the vertical grey bar.

Line 360-361: Sure it depends on "soil-intrinsic physical properties", but you never actually say what these are. Also, it also depends on the soil variables (soil temperature and soil moisture content)s, so please add this?

See our answer in the major comments section.

Line 369: replace "more dry" with "drier"

We replaced as suggested.

In Figure 6: Has the number of hours colouring code been explained properly and is it actually used much in the description around this graph? I found it a bit confusing. Around line 445 you discuss it a bit but it is not clear to me how these number of hours were counted or selected?

Thank you for pointing this out, indeed we forgot to add this information in the figure caption. We added it there and also clarified the information gain that we explain around line 445.:

*"One could argue that the sum of negative fluxes over shorter (less observations per night) and longer (more observations per night) timespans creates an artificial linear relationship between the two measurement instruments. In this case, however, the linear relationship between $F_{IN,EC}$ and $F_{IN,LYS}$ would break when considering only a specific timespan. We find, however, also a strong linear relationship when considering time periods of only one specific length (i.e. four hours: R = 0.6). This indicates that for continuous measurements of $F_{IN,EC}$ a substantial part cannot be (solely) explained by noise."*

Line 435-436: I don't know what you mean by "This is not surprising given that under stable nighttime conditions F is suspected to leave the control volume other than in the vertical direction (advection, drainage flows) and thus undetected by the EC sensor". Is there a word missing ("not" in front of "suspected")?

We understand that the sentence expressed what we meant in a very roundabout way. We have rephrased it and hope that it is now clearer:

*"Nevertheless, a structural difference between the measuring instruments in the form of a bias remains. This is not surprising as under stable nighttime conditions the ratio between vertical and non-vertical (drainage and advection) movement of F is expected to be smaller. As a result, a larger proportion of the total F leaves the source area undetected by the EC sensor than in daytime measurements with good atmospheric mixing (Wohlfahrt et al., 2005)."*

Following on from this: You have not defined control volume, footprint, fetch etc. higher up. In fact a footprint analysis for the data would be very welcome.

*Please be referred to our answers in the major comments section.*

Around Figure 8. The discussion here focusses on the temporally variable soil variables Ts and SWC. But it is not more likely that modest variations in clay content (and SOM?) among these lysimeters affect the adsorptive capacity of the soil in these lysimeters? Verhoef et al. (and Kosmas?) showed that adsorption is quite sensitive to clay content?

This comment has been addressed within the answer to the mayor comments section by adding the following sentence:

*"This means that the lumped effect of static properties which might deviate between lysimeters such as clay or organic matter carries a lower information content for the prediction of the differences between instruments."*

Around Figure 8. There is no real explanation/discussion of the SHAP plots c and d. I am not sure how to interpret these and what it all means.

Thank you very much for pointing this out. We realize that in the first version of the manuscript, the link between Figure 8 (the model results) and the conclusions we drew based on these results was often not clear. In the revised version, we have added the appropriate references to the figure panels in the respective sentences and hope that the connections between the Figure panels and their interpretation are clear now.

Lines 576 and 577: incorporating vapour fluxes is not just important for hydrological models but also for land surface models. It would be good to make that point here and add the following reference? Garcia Gonzalez, R. , Verhoef, A. , Vidale, P. , Braud, I. (2012) Incorporation of water vapour transfer in the JULES Land Surface Model: implications for key soil variables and land surface fluxes. Water Resources Research, 48 (5). ISSN: 0043-1397 | doi: https://dx.doi.org/10.1029/2011WR011811

Thank you for making us aware of this interesting read. We added the reference to the sentence as suggested.

---

## Author Comment (AC2)

| DOI | https://doi.org/10.5194/egusphere-2023-2556 |
|---|---|
| Interpretability of negative latent heat fluxes from Eddy Covariance measurements during dry conditions | |
| Author(s) | Sinikka J. Paulus et al. |
| Handling Editor | Andrew Feldman |
| Manuscript type | Research article |
| Status | Final response (author comments only) |

**Response to Report from Reviewer #2**

Reviewer comments are printed in black.
Answers are printed in blue below the respective comment.
* * *
My congratulations and thanks to the authors for putting together a clear and compelling case for detecting soil water vapor adsorption (SVA) using eddy covariance measurements. I found the paper easy to follow with interesting results and conclusions that were well supported.

Thank you very much for this kind assessment of our work

I have only some paper specific edits to suggest and comments that need clarification (minor revisions to address before publication).

L16-17. Wouldn't this claim about the prevalence of SVA be better supported by using the lysimeter results themselves? If so, change the text and move this above the results on how well EC does at detecting these events.

Thank you for your suggestion on the abstract structure. Since our study focuses on the interpretability of the negative latent heat flux (through the comparison to lysimeters), and to a lesser extent to the SVA measurement of the lysimeters, we decided that keeping the original structure of the abstract better reflects the manuscript's content.

L 20. Here and in the main body I found this result on using RF to explain mismatches between the two techniques to be confusing. In the text, I thought you were using RF mainly to explain mismatches between lysimeters themselves rather than differences between Fin estimates between EC and lysimeter?

Thank you for pointing out the ambiguity in the explanation of the conclusions of our model.

The model was designed to predict the mismatch between the difference between the lysimeter and the Eddy Covariance observations - which is why in Line 20 (abstract) we wrote "Based on a random Forest feature selection we found the mismatch between the EC vs lysimeter results".

We, therefore, addressed this comment in the respective results section 4.4 by revising the respective paragraph and clarifying the sentence, which was potentially misleading (line 468):

*"We investigated the potential reasons for the mismatch between the two measurement methods by means of a predictor variable selection procedure followed by a random forest model analysis with the deviation between EC and lysimeter as the dependent variable (Jung & Zscheischler, 2013). "*

However, we think that generally the use of the RF is clearly explained since we state right at the beginning of the relevant paragraph (line 476): *"The primary factor influencing the variation between instruments is SWC within the lysimeters. The deviation between instruments decreases at lower SWC (Fig 7c) and higher Ts (Fig 7d)."*

However, it is of course very important to us that readers can understand the result of the model. In case that also after a second read this section is still unclear to the reviewer, we would revise the text again.

L30. Is there a citation for this being an underrepresented component of the research?

Yes, you are right that having a citation here is appropriate. We added the respective citation:

Saaltink, M. W., Kohfahl, C., & Molano-Leno, L. (2020). Analysis of water vapor adsorption in soils by means of a lysimeter and numerical modeling. *Vadose Zone Journal*, *19*(1), e20012. https://doi.org/10.1002/vzj2.20012

L91. "distance between the sampling height and the ground"

We received a very similar suggestion from reviewer 1 and decided to add his/her suggestion: "the vertical distance between the EC sensors and the adsorbing soil surface"

L100. Suggest using "forces" instead of force fields.

We adapted the wording.

L108. "escape the liquid phase"

We adapted the wording.

Figure 1. Great summary figure!

Thank you very much, we are pleased to see that the figure is well received.

L71. "…integral turbulence characteristics were removed" - This wasn't mentioned for the other site. Also, is this based on measurements of turbulent mixing strength like u*?

Thank you for pointing out the differences in the depth of the description of the Eddy Covariance data processing. The filtering for time periods based on the test on integral turbulence characteristics was performed at both sites, following the method described in

Foken and Wichura (1996). We added the respective information to the processing description in ES-LMa:

*"Standard integral turbulence characteristics were identified and most problematic records removed (Foken and Wichura, 1996)."*

Apart from this detail, we decided to add the information that the two softwares used (EddyPro in ES-LMa and TK3 in DE-RuS) show close agreement (Fratini and Mauder 2014) to make more clear that there is no substantial difference in the processing of the EC flux data between the two sites:

*"The two softwares used to process the raw data at the two sites (EddyPro and TK3) have been shown to be in good agreement (Fratini and Mauder, 2014)."*

Foken, Th. and Wichura, B.: Tools for quality assessment of surface-based flux measurements, Agric. Forest Meteorol., 78, 83-105, https://doi.org/10.1016/0168-1923(95)02248-1

Fratini, G. and Mauder, M.: Towards a consistent eddy-covariance processing: an intercomparison of EddyPro and TK3, Atmos. Meas. Tech., 7, 2273–2281, https://doi.org/10.5194/amt-7-2273-2014, 2014.

L277.  $U* = 0.01$ m/s is extremely low. Is this a typo, maybe 0.1?

This is not a typo, but an important point that the reviewer raises here. After discussion with our co-authors, we decided to re-run our entire analysis based on a standard EC post-processing $u*$ - threshold that is site-specific and seasonally dynamic.

We changed the (former) line 277 accordingly to:

*"However, to be conservative we determined periods with low turbulent mixing based on $u*$ thresholds for each season of a site year. The effect of the $u*$ on the agreements between the two measurement methods was evaluated by removing measurements below the established threshold. To take into account the uncertainty of the $u*$ threshold estimate, this was repeated for the 5th and 95th percentile of the $u*$ threshold estimate (Papale et al. 2006; Wutzler et al. 2018, thresholds given in table F1)."*

The $u*$ thresholds (5th, 50th, and 95th percentile) vary between 0.050 and 0.103 m s$^{-1}$ in ES-LMa* and 0.054 and 0.177 m s$^{-1}$ in DE-RuS. The change in the $u*$ threshold affected the results reported in Section 4.2 and 4.3 including the respective tables and figures (Table 2, Table 3, Fig. 5, Fig. 6). In summary, the threshold confirmed that the flux direction between the two measurement methods is nearly constant independent of the selected $u*$ threshold. The comparison of the mean nighttime flux magnitude revealed that there is a 25 % reduction in the error (MAE) during conditions of good turbulence, compared to including the low turbulence conditions (in the preprint). The correlation between instruments increased from 0.66 to 0.79 in ES-LMa*. However, with this threshold an even lower number of observations in DE-RuS remains (only 4 nights when both methods measure $F_{IN}$ under turbulent conditions). Therefore, we refrain from interpreting the statistics at the temperate

site but keep the (updated) results in Appendix H for completeness. The model analysis, was kept as it was in the preprint with an u* threshold of only 0.01 m s$^{-1}$: This way we test the dependence of the half-hourly mismatch between the two measurement methods of the u* magnitude in a data-driven way. We added a sentence about the possible reason of the u* being less important than the rest of the variables:

*"It is therefore possible that soil heterogeneity conceals the effect of variables associated with EC uncertainty on the mismatch, which should be checked in a more homogeneous ecosystem. This is supported by the detectable effect of the u* that shows that the discrepancy between instruments decreases with higher u* (see Figure J1), but its effect on the mismatch is one order of magnitude smaller than the effect of lysimeter SWC and Ts."*

In addition to the changes in the results section and the respective figures and tables, the following new tables and figures were added to the appendix section:

- Table F1: reporting the different u* thresholds at both sites,
- Table F2: showing the effect of different u* thresholds on the flux direction agreement
- Table F3: reporting the statistics for the comparison between $F_{IN,EC}$ and $F_{IN, LYS}$ across the range of u* thresholds (5%; 95%)
- Figure J1: influence of single observations on the mismatch between EC and lysimeters (SHAP marginal plot; similar to the ones shown in Figure 8 (c) and (d) )

Figure 3. This is very compelling evidence of EC detection of negative fluxes indicating SVA. Would it also be possible to select a few multiday periods with substantial Ein and show mean diurnal plots of E for both the lysimeters and the flux tower? Just 2 or 3 of these would show whether there is strong evidence of SVA during dry periods and its magnitude relative to the daytime values. I know this is quantified later in the manuscript but this would be a way to show it visually.

Thank you for your suggestion. We agree with this idea.

We have decided to illustrate the diel measurements using four days of data in August 2019. Although the $F_{IN,EC}$ cannot really be called substantial, we think it illustrates well that almost all $F_{EC}$ values were negative during the nights, all lysimeters record weight increases ($F_{IN,LYS}$) and the RH never rises above 75%, showing that there is no formation of dew or fog. We added the figure as an Appendix Figure.

[Figure]

*Diurnal measurements of water vapor flux with (a) eddy-covariance ($F_{EC}$) and (b) the five lysimeters ($F_{LYS}$); the color code shows the respective station L1, L2, L3, L5, and L6) at ES-LMa\* from 11.08.2019 18:00 h until 15.08.2019 18:00. Panel (c) illustrates the course of relative humidity (RH) at 2 m height above the soil surface. Black vertical lines illustrate sunset and sunrise (determined by the geographic coordinates of the field site).*

L360. I don't understand "the process is independent of climate". It seems highly dependent on climate conditions. Please clarify.

The reviewer is right that the frequency of the SVA occurrence depends on climate conditions, as it clearly depends on meteorological conditions.

In paragraph proceeding l.360, however, we showed that SVA occurs also in the temperate climate during conditions of climate extremes. Therefore, here we wanted to point out that the process itself can happen in different climatic areas (as long as soil water content is low and RH high) because it is a soil intrinsic physical property (while climate is defined only by the mean conditions and not constant over time). We changed the sentence accordingly and hope that this point becomes more clear now from the text:

*"It underscores that while the probability of occurrence of SVA is influenced by climate (i.e. more common in semi-arid and arid regions), it can also occur in more humid regions. This is because it depends on soil-intrinsic physical properties, such as texture (clay content, clay mineralogy, and organic carbon content) (Orchiston, 1956; Arthur et al. 2019, Yukselen Akoy, 2010), soil structure that affects vapor transport characteristics (i.e. soil diffusion coefficient), and can happen anywhere if the dynamic requirements like temperature and moisture gradients are met."*

Table 2. Could you please add the number of periods/days that the lysimeters recorded these events?

This number is given in the first line of the table (n nights) - since as described in the manuscript, the process occurs nearly exclusively at night in our ecosystem.

L434. Why better? Couldn't it be that the lysimeter spatial mean is biased high compared to the broader EC scale?

In the respective paragraph, we don't state that the lysimeter measurements are more accurate but state that the statistical metrics for the comparison between the median across lysimeter columns and the Eddy covariance instrument are better than between EC and individual columns. We hypothesize in sentence L.434, that this might be because the median across lysimeters better represents the spatial mean (since each lysimeter only covers a small spatial scale of 1 m²).

We hope that we could clarify this in the revised version of the text and by adding the footprint climatology following the suggestion of reviewer 1. It now states:

*"However, we find higher agreement between EC and the median across the lysimeters (Table 3) than between EC and individual lysimeters (Table H1). One interpretation of this result could be that each lysimeter covers a smaller spatial scale (1 m² each) compared to the EC (illustrated in Figure 2 as footprint climatology) but the average across lysimeters is a better representation of the spatial mean and is therefore more in line with the EC observations."*

L484-486. I had trouble understanding this conclusion. I thought the sentences above were talking about what explains the differences between lysimeters and not between EC and lysimeters. Can you please explain and maybe rewrite this sentence?

This comment was already addressed in our answer to the comment related to line 20, (which refers to the summary of our model results in the abstract).

Figure 9 . Both axes in (c) are labeled as IN.

Thank you very much for spotting this. We changed the respective label.

L519-523. I think this text applies to bare soil evaporation. Evaporation from mixed plant/soil conditions as you have for the lysimeters and EC fluxes occurs differently than this in that stage 2 isn't just about diffusion-limited processes through the soil as plant transpiration is also a substantial element for this period (or at least for all but the driest conditions).

The reviewer makes a valid point for ecosystems with active vegetation. In our ecosystem, the grasses and hence - the vegetation in the EC footprint and on the lysimeters, have withered and don't transpire since they are dead during the focus period. This condition is described in the section site description (3.1) and again in the first paragraph of the results section (4.1). The soil is not bare since there is still plant residuals left, but transpiration is not part of the soil-atmosphere vapor exchange in this period (until it rains, which is also when SVA ceases due to rising SWC).

Therefore, we argue that the conditions that the discussion is based on are met but we've added the valid constraint that the reviewer pointed out ("in the absence of transpiration") to sentence line 518.

---

## Author Response (AR1)

| DOI | https://doi.org/10.5194/egusphere-2023-2556 |
|---|---|
| Interpretability of negative latent heat fluxes from Eddy Covariance measurements during dry conditions | |
| Author(s) | Sinikka J. Paulus et al. |
| Handling Editor | Andrew Feldman |
| Manuscript type | Research article |
| Status | Final response (author comments only) |

**Response to Report from Editor**

We've received comments from two reviewers, both experts in this field. Both are supportive and recommend the manuscript for publication after minor revisions. After reading myself, I agree with the two reviewers that the paper is well written and interesting.

Thank you very much for this kind assessment of our work.

Some brief comments on the reviewer's points: While I can see Reviewer 1's point, I agree with the authors' response that it is good to keep the wetter site in Germany in the analysis to show the contrast with the drier site. I also agree with the second reviewer that some more motivation and discussion of explaining the mismatch between EC and lysimeter is interesting rather than explaining the lysimeter patterns.

I also have some additional comments below to consider. Ultimately, we invite the authors to implement their revisions.

1)      The condition for SVA discussed in lines 230-240 might need some more discussion. Specifically, if the soil temperature (Ts) is below the dew point, wouldn't this be conditions for dew formation on the soil surface/pores? Therefore, SVA in the soil occurs below the dew point? Dew should form when the air temperature (or the bare soil surface Ts in this case) drops below the dew point. If I am misunderstanding, please provide some more detail on this point.

Thank you very much for spotting this error! The partitioning equations for dew and SVA at the end of Page 9 (230) was wrong. It should be
"$Tdew_{0.1m} > T_s$" for dew
"$Tdew_{0.1m} < T_s$" for SVA
(Ts stands for surface temperature in our manuscript) which is in line with the definition that you refer to in this comment.

2)      I and g are not defined in the caption in Fig. 1

Thank you, we added the following explanation into the caption of Figure 1:

*The representations (a and b) illustrate that at constant atmospheric rH of 60 % at a temperature of 20 °C, the vapor flux direction and phase change **(I and g for liquid and gas)** within the soil are opposite for different soil water potentials.*

3)      Consider showing a time series to demonstrate SVA with the lysimeter and EC. Currently, while detailed, Fig. 3 and 4 are quite dense and potentially difficult for some readers to see the SVA process. It looks like Reviewer 2 requested something like this.

We have indeed added an additional figure to the appendix (Figure D1) showing the course of diel measurements from EC and lysimeters, as suggested by reviewer 2, and decided to add surface temperature and dew point temperature to show that the surface is too hot for dew formation during this period.

[Figure]

**Figure D1.** Diurnal measurements of water fluxes from (a) eddy-covariance ($F_{EC}$) and (b) the five lysimeters (L1, L2, L3, L5, and L6) from 11.08.2019 18:00 h until 15.08.2019 18:00 at ES-LMa*. Panel (c) illustrates the course of relative humidity (RH) at 2 m height above the soil surface together with surface ($T_s$) and dewpoint temperature ($T_{dew}$). Black vertical lines illustrate sunset and sunrise (determined by the geographic coordinates of the field site).

4)      An error propagation analysis was mentioned around line 442. Please consider showing this analysis in the supplemental materials given the value it adds.

Thank you for this suggestion. We have added a figure to the Appendix section (Figure J1) showing the relative proportion of the error-propagated random error compared to the total sum of negative LE fluxes, which reinforces the argument we make around line 442.

**Appendix J: Proportion of random error on $F_{IN,EC}$**

[Figure]

**Figure J1.** The relative random error shown on the y-axis is the proportion of the random error from the total inward flux measurements $F_{IN,EC}$ of the EC for each nights. The half hourly $F_{IN,EC}$ measurements per night were summed. The random error per night was determined by propagating the random error of the half houly measurements using standard deviations.

-Andrew Feldman

Thank you very much for contributing and helping us to improve this manuscript.

Sinikka Paulus